# RETHINKING UNCERTAINTY ESTIMATION IN LLMS: A PRINCIPLED SINGLE-SEQUENCE MEASURE

**Lukas Aichberger**[1,*], **Kajetan Schweighofer**[1,*], **Sepp Hochreiter**[1,2]

[1] ELLIS Unit Linz and LIT AI Lab, Institute for Machine Learning,
  Johannes Kepler University Linz, Austria
[2] NXAI GmbH, Linz, Austria
[*] Joint first authors
 `{aichberger, schweighofer, hochreit}@ml.jku.at`

## ABSTRACT

Large Language Models (LLMs) are increasingly employed in real-world applications, driving the need to evaluate the trustworthiness of their generated text. To this end, reliable uncertainty estimation is essential. Leading uncertainty estimation methods generate and analyze multiple output sequences, which is computationally expensive and impractical at scale. In this work, we inspect the theoretical foundations of these methods and explore new directions to enhance computational efficiency. Building on the framework of proper scoring rules, we find that the negative log-likelihood of the most likely output sequence constitutes a theoretically principled uncertainty measure. To approximate this alternative measure, we propose `G-NLL`, obtained using a single output sequence from greedy decoding. This approach streamlines uncertainty estimation while preserving theoretical rigor. Empirical results demonstrate that `G-NLL` achieves state-of-the-art performance across various scenarios. Our work lays the theoretical foundation for efficient and reliable uncertainty estimation in natural language generation, challenging the necessity of the prevalent methods that are more complex and resource-intensive.

## 1 INTRODUCTION

Despite advances in natural language generation (NLG), determining the trustworthiness of generated text remains challenging. Addressing this requires reliably estimating the uncertainty a language model has regarding its generated text. Although a low level of uncertainty does not guarantee factual correctness, particularly when the generated text is based on consistent but inaccurate training data, uncertainty estimates remain a reliable indicator of errors at present (Farquhar et al., 2024).

Assessing predictive uncertainty in Large Language Models (LLMs) is challenging due to their stochastic, autoregressive nature. Each token is selected probabilistically, leading to diverse outputs for the same input. Furthermore, the vast space of possible sequences is computationally intractable. Common uncertainty estimation methods thus rely on expectations over output distributions, such as sequence entropy (Malinin and Gales, 2021; Kuhn et al., 2023; Duan et al., 2024; Farquhar et al., 2024), which in turn requires sampling multiple output sequences. However, this is computationally costly due to the large number of model parameters. As a result, only a small subset of outputs is sampled in practice. However, differences between sampled sequences do not always indicate uncertainty, as they may vary lexically while remaining semantically similar. Some methods use inference models to assess semantics (Kuhn et al., 2023; Aichberger et al., 2025; Farquhar et al., 2024), improving uncertainty estimates but adding complexity and additional computation. These challenges make large-scale uncertainty estimation impractical for real-world applications.

Efficient uncertainty estimation methods are needed to ensure the trustworthiness of the language model's answer without imposing excessive computational demands. To address this need, we theoretically motivate that uncertainty measures require only a single output sequence. We do so by building on insights from the framework of proper scoring rules (Gneiting and Raftery, 2007) that has recently been investigated for uncertainty estimation in the standard classification setting (Kotelevskii et al., 2025; Hofman et al., 2024). Specifically, we extend proper scoring rules for uncertainty

estimation to NLG and explore the zero-one score as an alternative to the prevalent logarithmic score. The resulting uncertainty measure is straightforward: it is the negative log-likelihood of the most likely output sequence. Importantly, the measure is based on the most likely output sequence, rather than an arbitrary one. However, determining the most likely output sequence remains computationally expensive. Therefore, we propose `G-NLL`, an approximation that is maximally efficient and minimizes algorithmic complexity, while maintaining state-of-the-art performance.

In parallel, recent work has considered the likelihood of a single output sequence for uncertainty estimation in NLG (Fadeeva et al., 2023; 2024; Vazhentsev et al., 2024; Abbasi-Yadkori et al., 2024; Vashurin et al., 2025a;b). Notably, Fadeeva et al. (2023) introduce the maximum sequence probability (MSP) as a baseline, which corresponds to the negative log-likelihood of the most likely output sequence. However, their ad hoc formulation does not provide a theoretical justification or a discussion on how to best approximate it. More generally, prior work has not properly characterized the MSP as a measure for uncertainty in NLG. As a result, prior work often relies on unfavorable length-normalization (Qiu and Miikkulainen, 2024; Chen et al., 2024; Bakman et al., 2024; Yaldiz et al., 2025), does not focus on the *most likely* output sequence, or even overlooks single-sequence measures entirely as baselines (Malinin and Gales, 2018; Manakul et al., 2023; Kuhn et al., 2023; Farquhar et al., 2024; Nikitin et al., 2024; Kossen et al., 2024; Duan et al., 2024).

To close this gap, we are the first to provide a theoretical justification for the MSP as a principled, single-sequence measure of uncertainty in NLG. We examine the properties of this alternative uncertainty measure and compare it to established measures from both theoretical and empirical perspectives. Additionally, we propose `G-NLL` as an efficient approximation of the MSP and analyze why sampling-based or length-normalized alternatives are detrimental to its approximation quality. Our experiments demonstrate that `G-NLL` matches and even exceeds the estimation quality of established methods across various model classes, model sizes, training stages, tasks, datasets, and evaluation metrics. By maintaining theoretical rigor, `G-NLL` offers an effective and scalable approach to uncertainty estimation in NLG. It serves not only as a strong baseline for future work, but also as a practical solution for deploying uncertainty estimation in LLMs across real-world applications.

To summarize, our main contributions are as follows:

- We derive the negative log-likelihood of the most likely output sequence (i.e., the MSP) as a measure of uncertainty in NLG, building on established principles from uncertainty estimation theory and proper scoring rules.

- We provide a theoretical analysis of the MSP and existing uncertainty measures, and show that estimating this single-sequence measure possesses desirable properties for practical scenarios.

- We propose `G-NLL` as an effective approximation of the MSP, and demonstrate that it empirically outperforms state-of-the-art methods while significantly reducing computational costs.

## 2 Predictive Uncertainty in NLG

We begin by reviewing language modeling to formalize predictive uncertainty in NLG. Sec. 2.1 introduces proper scoring rules and their connection to predictive uncertainty, and Sec. 2.2 revisits established measures within this framework. Sec. 2.3 then introduces the maximum sequence probability and its approximation `G-NLL` under an alternative scoring rule.

**Preliminaries.** We assume a fixed training dataset $\mathcal{D} = \{s_i\}_{i=1}^N$ consisting of $N$ token sequences $s = (s_1, ..., s_\tau)$ where individual tokens $s_t \in \mathcal{V}$ are from a given vocabulary $\mathcal{V}$. Each token at step $t$ is assumed to be sampled according to the predictive distribution $p(s_t \mid s_{<t}, w^*)$, conditioned on the sequence of preceding tokens $s_{<t}$ and the true (but unknown) language model parameters $w^*$. We assume that the given model class can theoretically represent the true predictive distribution, a common and usually necessary assumption (Hüllermeier and Waegeman, 2021). The likelihood of some model parameters $\tilde{w}$ matching $w^*$ is given by the posterior $p(\tilde{w} \mid \mathcal{D}) = p(\mathcal{D} \mid \tilde{w})p(\tilde{w})/p(\mathcal{D})$.

The input to a given language model parameterized by $w$ is a sequence $x = (x_1, ..., x_M)$ and the output is a sequence $y = (y_1, ..., y_T) \in \mathcal{Y}_T$, with $x, y \in \mathcal{V}$ and $\mathcal{Y}_T$ being the set of all possible output sequences with sequence length $T$. The likelihood of a token $y_t \in y$ being generated by the language model is conditioned on both the input sequence and all previously generated tokens, denoted as $p(y_t \mid x, y_{<t}, w)$. The likelihood of output sequences $y \in \mathcal{Y}_T$ being generated by

the language model is then the product of the individual token probabilities, which is denoted as $p(\boldsymbol{y} \mid \boldsymbol{x}, \boldsymbol{w}) = \prod_{t=1}^{T} p(y_t \mid \boldsymbol{x}, \boldsymbol{y}_{<t}, \boldsymbol{w})$ (Sutskever et al., 2014), while the heuristic length-normalized variant is $\bar{p}(\boldsymbol{y} \mid \boldsymbol{x}, \boldsymbol{w}) = \exp\left\{ \frac{1}{T} \sum_{t=1}^{T} \log p(y_t \mid \boldsymbol{x}, \boldsymbol{y}_{<t}, \boldsymbol{w}) \right\}$ (Malinin and Gales, 2021).

Calculating the likelihood that a specific output sequence is generated by the language model parameterized by $\boldsymbol{w}$ is straightforward. The language model directly provides the token likelihoods for a given input sequence. However, determining the full probability distribution on all possible output sequences is considerably more challenging, since the size of $\mathcal{Y}_T$ increases exponentially with the sequence length. The computational complexity of evaluating all possible output sequences increases with $\mathcal{O}(|\mathcal{V}|^T)$. Since the vocabulary sizes $|\mathcal{V}|$ of modern LLMs are well over a hundred thousand tokens, this distribution becomes intractable to determine, even for relatively short maximal sequence lengths $T$ (Dubey et al., 2024).

### 2.1 PROPER SCORING RULES AND THE RELATION TO UNCERTAINTY MEASURES IN NLG

We next give an introduction to proper scoring rules and discuss how they give rise to uncertainty measures. For more details, in the standard classification setting, we refer to Hofman et al. (2024); Kotelevskii et al. (2025). Proper scoring rules are a class of functions that evaluate the quality of probabilistic predictions by assigning a numerical score based on the predictive distribution and actual observations (Gneiting and Raftery, 2007). In particular, a proper scoring rule is an extended real-valued function $S : \mathcal{P} \times \mathcal{Y}_T \to [-\infty, \infty]$, such that $S(p, \boldsymbol{y})$ is $\mathcal{P}$-quasi-integrable over a convex class of probability measures $\mathcal{P}$. A scoring rule is called proper relative to $\mathcal{P}$ if the expected score is minimized when the evaluated distribution $p \in \mathcal{P}$ coincides with the distribution from which the outcomes $\boldsymbol{y} \in \mathcal{Y}_T$ are sampled from, and it is called strictly proper if this minimum is unique. In the context of uncertainty estimation in NLG, proper scoring rules assign a numerical value that reflects how much probability the predictive distribution of the *true* model $p(\boldsymbol{y} \mid \boldsymbol{x}, \boldsymbol{w}^*)$ places on an observed output sequence $\boldsymbol{y}'$, denoted as $S(p(\boldsymbol{y} \mid \boldsymbol{x}, \boldsymbol{w}^*), \boldsymbol{y}')$. Here, the output sequence $\boldsymbol{y}'$ is an independent notational copy of $\boldsymbol{y}$ for clarity, where necessary. In the following, scoring rules are used in the loss convention, so larger expected scores correspond to higher predictive uncertainty.

To obtain concrete uncertainty measures, we need to make two specific assumptions (Schweighofer et al., 2025). First, we have to define the predictive distribution used to sample output sequences. Following Aichberger et al. (2025), we we use a single, *given* language model with parameters $\boldsymbol{w}$ to sample output sequences $\boldsymbol{y}' \sim p(\boldsymbol{y}' \mid \boldsymbol{x}, \boldsymbol{w})$. This assumption is also used in other works (Kuhn et al., 2023; Fadeeva et al., 2024; Farquhar et al., 2024) and is intuitively reasonable, since our main concern is the uncertainty of the output of a specific language model. Thus, we consider the expected score for possible output sequences under the predictive distribution of the given language model, denoted as $\mathrm{E}_{p(\boldsymbol{y}'|\boldsymbol{x}, \boldsymbol{w})}[S(p(\boldsymbol{y} \mid \boldsymbol{x}, \boldsymbol{w}^*), \boldsymbol{y}')]$. This quantifies how well the predictive distribution of the given language model aligns with the true predictive distribution, capturing predictive uncertainty. Second, we have to define how the true model is approximated. We consider a Bayesian approximation of the true model, i.e., each possible language model $\tilde{\boldsymbol{w}}$ according to its posterior probability $p(\tilde{\boldsymbol{w}} \mid \mathcal{D})$ (Schweighofer et al., 2023b;a). Thus, we perform a posterior expectation over the expected score $\mathrm{E}_{p(\tilde{\boldsymbol{w}}|\mathcal{D})}\left[\mathrm{E}_{p(\boldsymbol{y}'|\boldsymbol{x}, \boldsymbol{w})}[S(p(\boldsymbol{y} \mid \boldsymbol{x}, \tilde{\boldsymbol{w}}), \boldsymbol{y}')]\right]$, which can be additively decomposed into an entropy and a divergence term (Gneiting and Raftery, 2007; Kull and Flach, 2015):

$$\underbrace{\mathrm{E}_{p(\tilde{\boldsymbol{w}}|\mathcal{D})}\Big[\mathrm{E}_{p(\boldsymbol{y}'|\boldsymbol{x}, \boldsymbol{w})}\big[S\big(p(\boldsymbol{y} \mid \boldsymbol{x}, \tilde{\boldsymbol{w}}), \boldsymbol{y}'\big)\big]\Big]}_{\text{expected score}} = \tag{1}$$

$$\underbrace{\mathrm{E}_{p(\boldsymbol{y}'|\boldsymbol{x}, \boldsymbol{w})}\big[S\big(p(\boldsymbol{y} \mid \boldsymbol{x}, \boldsymbol{w}), \boldsymbol{y}'\big)\big]}_{\text{entropy term}} + \underbrace{\mathrm{E}_{p(\tilde{\boldsymbol{w}}|\mathcal{D})}\Big[\mathrm{E}_{p(\boldsymbol{y}'|\boldsymbol{x}, \boldsymbol{w})}\big[S\big(p(\boldsymbol{y} \mid \boldsymbol{x}, \tilde{\boldsymbol{w}}), \boldsymbol{y}'\big) - S\big(p(\boldsymbol{y} \mid \boldsymbol{x}, \boldsymbol{w}), \boldsymbol{y}'\big)\big]\Big]}_{\text{divergence term}}.$$

The expected score over possible output sequences $\boldsymbol{y}'$ and language models $\tilde{\boldsymbol{w}}$ captures the *total* uncertainty of the *given* language model. The entropy term reflects *aleatoric* uncertainty, which quantifies the inherent stochasticity of generating output sequences with a given language model (Aichberger et al., 2025). The divergence term reflects *epistemic* uncertainty, which quantifies the uncertainty due to lack of knowledge about the true language model parameters arising from limited data (Houlsby et al., 2011; Gal, 2016; Malinin, 2019; Hüllermeier and Waegeman, 2021).

The remaining degree of freedom is the choice of a proper scoring rule $S(\cdot, \cdot)$, which determines the concrete form of the uncertainty measures. In the following, we consider two canonical proper scoring rules: the logarithmic score $S_{\log}(\cdot, \cdot)$ in Sec. 2.2 and the zero-one score $S_{\text{0-1}}(\cdot, \cdot)$ in Sec. 2.3.

## 2.2 ESTABLISHED UNCERTAINTY MEASURES IN NLG BASED ON THE LOGARITHMIC SCORE

The logarithmic score is typically assumed implicitly in both the standard classification (Houlsby et al., 2011; Gal, 2016) and the NLG setting (Malinin and Gales, 2021; Kuhn et al., 2023) to derive uncertainty measures. This is due to the grounding of the resulting uncertainty measures in information theory (Lahlou et al., 2023; Gruber and Buettner, 2023; Hofman et al., 2024; Kotelevskii et al., 2025). In the context of NLG, the logarithmic score considers the negative log-likelihood of a generated output sequence $\boldsymbol{y}'$:

$$\mathrm{S}_{\log}\big(p(\boldsymbol{y} \mid \boldsymbol{x}, \cdot), \boldsymbol{y}'\big) \ = \ -\log p(\boldsymbol{y} = \boldsymbol{y}' \mid \boldsymbol{x}, \cdot) \,. \tag{2}$$

Substituting the logarithmic score into Eq. (1) results in the total uncertainty being the cross-entropy $\mathrm{CE}(\cdot \,;\, \cdot)$ between the output sequence distribution of the given model and those induced by models integrated over the entire posterior $p(\tilde{\boldsymbol{w}} \mid \mathcal{D})$ (Aichberger et al., 2025) (see Apx. A.1 for details):

$$\underbrace{\mathrm{E}_{p(\tilde{\boldsymbol{w}}|\mathcal{D})}\big[\mathrm{CE}\big(p(\boldsymbol{y} \mid \boldsymbol{x}, \boldsymbol{w}) \,;\, p(\boldsymbol{y} \mid \boldsymbol{x}, \tilde{\boldsymbol{w}})\big)\big]}_{\text{total uncertainty}} \ = \tag{3}$$

$$\underbrace{\mathrm{H}\big(p(\boldsymbol{y} \mid \boldsymbol{x}, \boldsymbol{w})\big)}_{\text{aleatoric uncertainty}} \ + \ \underbrace{\mathrm{E}_{p(\tilde{\boldsymbol{w}}|\mathcal{D})}\big[\mathrm{KL}\big(p(\boldsymbol{y} \mid \boldsymbol{x}, \boldsymbol{w}) \,\big\|\, p(\boldsymbol{y} \mid \boldsymbol{x}, \tilde{\boldsymbol{w}})\big)\big]}_{\text{epistemic uncertainty}} \,.$$

The epistemic uncertainty is captured by a posterior expectation of the Kullback-Leibler divergence $\mathrm{KL}(\cdot \,\|\, \cdot)$ between the output sequence distribution of the given model and those induced by models across the entire posterior. This requires considering every possible parameterization of the model. Since modern LLMs have billions of parameters (Radford et al., 2018; Zhang et al., 2022; Dubey et al., 2024; Zuo et al., 2024), the epistemic uncertainty is challenging to estimate.

Thus, current work usually only focuses on the aleatoric uncertainty, captured by the Shannon entropy $\mathrm{H}(\cdot)$ of the output sequence distribution of the given language model (Kuhn et al., 2023; Aichberger et al., 2025; Farquhar et al., 2024). Computing this output sequence distribution still requires considering the entire set of possible output sequences $\mathcal{Y}_T$, which is intractable and has to be approximated, as discussed in the following.

**Predictive Entropy.** The aleatoric uncertainty under a given language model is the entropy of the output sequence distribution $p(\boldsymbol{y} \mid \boldsymbol{x}, \boldsymbol{w})$, commonly referred to as predictive entropy (PE) (Malinin and Gales, 2021). Intuitively, high PE implies that the language model is likely to generate different output sequences from the same input sequence, indicating high uncertainty. PE is generally estimated by Monte Carlo (MC) sampling of output sequences (Malinin and Gales, 2021):

$$\mathrm{H}\big(p(\boldsymbol{y} \mid \boldsymbol{x}, \boldsymbol{w})\big) \ = \ \mathrm{E}_{p(\boldsymbol{y}|\boldsymbol{x},\boldsymbol{w})}\big[-\log p(\boldsymbol{y} \mid \boldsymbol{x}, \boldsymbol{w})\big] \tag{4}$$

$$\approx \ \frac{1}{N} \sum_{n=1}^{N} -\log p(\boldsymbol{y}^n \mid \boldsymbol{x}, \boldsymbol{w}) \,, \qquad \boldsymbol{y}^n \sim p(\boldsymbol{y} \mid \boldsymbol{x}, \boldsymbol{w}) \,.$$

**Semantic Entropy.** Semantic entropy (SE) (Kuhn et al., 2023; Farquhar et al., 2024) is based on the fact that output sequences may be different on a token level but equivalent on a semantic level. In such cases, the PE can be misleading, as it indicates high uncertainty even when different output sequences have the same semantic meaning. Thus, instead of the entropy of the output sequence distribution, the entropy of the semantic cluster distribution is considered, denoted as $p(c \mid \boldsymbol{x}, \boldsymbol{w}) = \sum_{\mathcal{Y}_T} p(c \mid \boldsymbol{x}, \boldsymbol{y}, \boldsymbol{w}) \, p(\boldsymbol{y} \mid \boldsymbol{x}, \boldsymbol{w})$. The probability of an output sequence belonging to a semantic cluster is usually approximated with a separate natural language inference model. SE thus measures uncertainty about the semantics of output sequences and is defined as

$$\mathrm{H}\big(p(c \mid \boldsymbol{x}, \boldsymbol{w})\big) \ = \ \mathrm{E}_{p(c|\boldsymbol{x},\boldsymbol{w})}\big[-\log p(c \mid \boldsymbol{x}, \boldsymbol{w})\big] \tag{5}$$

$$\approx \ \frac{1}{N} \sum_{n=1}^{N} -\log p(c^n \mid \boldsymbol{x}, \boldsymbol{w}) \,, \qquad c^n \sim p(c \mid \boldsymbol{x}, \boldsymbol{w}) \,.$$

For details on how to construct a tractable approximation of SE, we refer to Aichberger et al. (2025).

**Discussion.** In general, each of these uncertainty measures is based on the logarithmic score and considers the distribution over all possible output sequences $p(\boldsymbol{y} \mid \boldsymbol{x}, \boldsymbol{w})$, which is defined over the entire set of possible output sequences $\mathcal{Y}_T$. Approximating expectations over this distribution requires sampling multiple output sequences from $\mathcal{Y}_T$, which is computationally expensive. In the following, we show that this requirement can be eliminated when considering an alternative proper scoring rule.

## 2.3 NEW UNCERTAINTY MEASURES IN NLG BASED ON THE ZERO-ONE SCORE

In principle, any proper scoring rule may be used to derive viable uncertainty measures (Kotelevskii et al., 2025; Hofman et al., 2024). Thus, we explore measuring predictive uncertainty based on the zero-one score instead of the logarithmic score. Although it has been considered in the standard classification setting, to the best of our knowledge, the zero-one score has not yet been considered as a proper scoring rule for deriving uncertainty measures in NLG. The zero-one score only considers the predictive distribution for the *most likely output sequence*:

$$\mathrm{S}_{0\text{-}1}\big(p(\boldsymbol{y} \mid \boldsymbol{x}, \cdot), \boldsymbol{y}'\big) = 1 - \mathbb{1}\big\{\boldsymbol{y}' = \underset{\boldsymbol{y} \in \mathcal{Y}_T}{\mathrm{argmax}}\, p(\boldsymbol{y} \mid \boldsymbol{x}, \cdot)\big\}. \tag{6}$$

Here, $\mathbb{1}\{\cdot\}$ denotes the indicator function, taking value $1$ if its argument is true and $0$ otherwise. Substituting the zero-one score into Eq. (1) results in the total uncertainty being the expected confidence that the given model assigns to the most likely output sequences of models across the entire posterior $p(\tilde{\boldsymbol{w}} \mid \mathcal{D})$ (see Apx. A.2 for details):

$$\underbrace{\mathrm{E}_{p(\tilde{\boldsymbol{w}}|\mathcal{D})}\big[1 - p(\boldsymbol{y} = \tilde{\boldsymbol{y}}^* \mid \boldsymbol{x}, \boldsymbol{w})\big]}_{\text{total uncertainty}} = \tag{7}$$

$$\underbrace{1 - p(\boldsymbol{y} = \boldsymbol{y}^* \mid \boldsymbol{x}, \boldsymbol{w})}_{\text{aleatoric uncertainty}} + \underbrace{p(\boldsymbol{y} = \boldsymbol{y}^* \mid \boldsymbol{x}, \boldsymbol{w}) - \mathrm{E}_{p(\tilde{\boldsymbol{w}}|\mathcal{D})}\big[p(\boldsymbol{y} = \tilde{\boldsymbol{y}}^* \mid \boldsymbol{x}, \boldsymbol{w})\big]}_{\text{epistemic uncertainty}},$$

where $\boldsymbol{y}^* = \mathrm{argmax}_{\boldsymbol{y} \in \mathcal{Y}_T}\, p(\boldsymbol{y} \mid \boldsymbol{x}, \boldsymbol{w})$ denotes the most likely output sequence under the given language model and $\tilde{\boldsymbol{y}}^* = \mathrm{argmax}_{\boldsymbol{y} \in \mathcal{Y}_T}\, p(\boldsymbol{y} \mid \boldsymbol{x}, \tilde{\boldsymbol{w}})$ the most likely output sequence under any language model parametrized by $\tilde{\boldsymbol{w}}$.

As in Eq. (3), the epistemic uncertainty is a posterior expectation that remains challenging to estimate. Following prior work, we again focus on the aleatoric uncertainty, which considers the likelihood of the most likely output sequence under the given language model. It turns out that the aleatoric uncertainty is equivalent to the maximum sequence probability (MSP) previously introduced in an ad hoc manner (Fadeeva et al., 2023; 2024).

While the aleatoric uncertainty derived from the logarithmic score requires approximating an expectation over the entire output sequence distribution by sampling multiple output sequences (see Eq. (4) and Eq. (5)), the one derived from the zero-one score only requires approximating the most likely output sequence under the given language model (see Eq. (7)). This distinction is crucial, as sampling multiple output sequences is computationally expensive, while approximating the most likely output sequence aligns directly with standard inference techniques widely used in LLMs.

For numerical stability, it is preferable to consider the logarithm of the aleatoric uncertainty in Eq. (7) and omit the constant offset of one, as it does not affect the relative ordering of uncertainty estimates. This yields the negative log-likelihood (NLL) of the most likely output sequence, which we denote as

$$\mathrm{M}\big(p(\boldsymbol{y} \mid \boldsymbol{x}, \boldsymbol{w})\big) := -\max_{\boldsymbol{y} \in \mathcal{Y}_T} \log p(\boldsymbol{y} \mid \boldsymbol{x}, \boldsymbol{w}) = -\max_{(y_1,\dots,y_T) \in \mathcal{Y}_T} \sum_t^T \log p(y_t \mid \boldsymbol{y}_{<t}, \boldsymbol{x}, \boldsymbol{w}) \tag{8}$$

$$\propto \underbrace{1 - p(\boldsymbol{y} = \boldsymbol{y}^* \mid \boldsymbol{x}, \boldsymbol{w})}_{\text{aleatoric uncertainty}} = 1 - \max_{(y_1,\dots,y_T) \in \mathcal{Y}_T} \prod_t^T p(y_t \mid \boldsymbol{y}_{<t}, \boldsymbol{x}, \boldsymbol{w}).$$

We note that this single-sequence measure is also known as the min-entropy, the most conservative of the Rényi-entropies (Rényi, 1961), which is a lower bound on PE in Eq. (4).

With this, the task of uncertainty estimation reduces to finding the most likely output sequence. However, the search space is the set of all possible output sequences $\mathcal{Y}_T$, so the exact computation remains intractable. To make the computation tractable, we propose replacing the full sequence maximization with a tokenwise maximization by moving the max operator inside the summation. This yields a very efficient approximation, since it simply corresponds to standard greedy decoding with the given language model. Thus, we can approximate the NLL of the most likely output sequence in Eq. (8) using greedy decoding (G), which we formally define as

$$\texttt{G-NLL} := -\sum_{t=1}^T \log\Big(\max_{y_t \in \mathcal{V}} p(y_t \mid \boldsymbol{x}, \boldsymbol{y}_{<t}, \boldsymbol{w})\Big) \approx \mathrm{M}\big(p(\boldsymbol{y} \mid \boldsymbol{x}, \boldsymbol{w})\big). \tag{9}$$

**Discussion.** Although uncertainty measures based on the logarithmic score could, in principle, perform well if the distribution over output sequences $p(\boldsymbol{y} \mid \boldsymbol{x}, \boldsymbol{w})$ — or equivalently the distribution over semantic clusters $p(\boldsymbol{c} \mid \boldsymbol{x}, \boldsymbol{w})$ — were tractable (as in standard classification tasks), these distributions are intractable for NLG due to the large vocabulary size and the auto-regressive nature of LLMs. As a result, sampling-based methods often yield crude approximations, suffering from computational costs and sampling variability. In contrast, G-NLL offers a principled alternative while eliminating the need for extensive sampling, making it highly practical and straightforward, while maintaining theoretical rigor through its foundation in proper scoring rules.

In general, uncertainty measures in NLG can be defined by a predictive distribution and a proper scoring rule: under the logarithmic score, the aleatoric uncertainty evaluated on $p(\boldsymbol{y} \mid \boldsymbol{x}, \boldsymbol{w})$ corresponds to PE (Eq. (4)), while evaluated on $p(c \mid \boldsymbol{x}, \boldsymbol{w})$ it corresponds to SE (Eq. (5)). Analogously, under the zero-one score, the aleatoric uncertainty evaluated on $p(\boldsymbol{y} \mid \boldsymbol{x}, \boldsymbol{w})$ corresponds to the maximum sequence probability (MSP), while evaluated on $p(c \mid \boldsymbol{x}, \boldsymbol{w})$ it yields the semantic counterpart of MSP, which we term the maximum cluster probability (MCP). G-NLL approximates MSP from a single output sequence, giving it a computational advantage over PE, whereas approximating MCP requires multiple output sequences, offering no such advantage over SE. We discuss this further in Apx. A.2. In the remainder of this paper, we focus on MSP and defer a detailed investigation of MCP to future work.

## 3 THEORETICAL ANALYSIS

Before empirically validating G-NLL in Sec. 5, we analyze the sample-complexity of approximating $\mathrm{H}(\cdot)$ and $\mathrm{M}(\cdot)$, showing that approximating the latter as done with G-NLL is desirable in the setting of LLMs. We consider a probability distribution of output sequences $p(\boldsymbol{y})$, dropping the dependency on $\boldsymbol{x}$ and $\boldsymbol{w}$ in this section for brevity. To incorporate importance sampling schemes, we consider access to a proposal distribution $q(\boldsymbol{y})$, e.g., by temperature-sampling from the LLM. We are interested in estimating $\mathrm{H}(p(\boldsymbol{y}))$ and $\mathrm{M}(p(\boldsymbol{y}))$ with their respective MC estimators:

$$\mathrm{M}\big(p(\boldsymbol{y})\big) = -\max_{\boldsymbol{y}} \log p(\boldsymbol{y}) \qquad\qquad \hat{\mathrm{M}}\big(p(\boldsymbol{y})\big) = -\max\big\{ \log p(\boldsymbol{y}^n)\big\}_{n=1}^N$$

$$\mathrm{H}\big(p(\boldsymbol{y})\big) = -\mathrm{E}_{q(\boldsymbol{y})}\left[\frac{p(\boldsymbol{y})}{q(\boldsymbol{y})} \log p(\boldsymbol{y})\right] \qquad\qquad \hat{\mathrm{H}}\big(p(\boldsymbol{y})\big) = -\frac{1}{N}\sum_{n=1}^N \frac{p(\boldsymbol{y}^n)}{q(\boldsymbol{y}^n)} \log p(\boldsymbol{y}^n)$$

where $\boldsymbol{y}^n \sim q(\boldsymbol{y})$. Furthermore, we assume boundedness, i.e., there exist constants $a, b$ such that $a \leqslant \log p(\boldsymbol{y}) \leqslant b$ due to logit clipping and non-zero softmax temperatures. Then the following holds:

**Theorem 1.** *Let $\log p(\boldsymbol{y}) \in [a,b]$, $\forall \boldsymbol{y} \in \mathcal{Y}_T$ and $\epsilon > 0$. Then, with probability $1 - \delta$, we get:*

1. *Sample-Complexity Bound for Maximum Log-Likelihood Estimation*

$$\left|\mathrm{M}\big(p(\boldsymbol{y})\big) - \hat{\mathrm{M}}\big(p(\boldsymbol{y})\big)\right| \leqslant \epsilon \quad \Longleftarrow \quad N \geq \frac{C_\epsilon}{P_\epsilon} \log\left(\frac{1}{\delta}\right), \tag{10}$$

   *where $\mathcal{Y}_\epsilon = \big\{\boldsymbol{y} \in \mathcal{Y}_T \mid \log p(\boldsymbol{y}^*) - \log p(\boldsymbol{y}) \leqslant \epsilon\big\}$,*
   *$P_\epsilon = \sum_{\boldsymbol{y}\in\mathcal{Y}_\epsilon} p(\boldsymbol{y})$, $Q_\epsilon = \sum_{\boldsymbol{y}\in\mathcal{Y}_\epsilon} q(\boldsymbol{y})$, and $C_\epsilon = P_\epsilon/Q_\epsilon$.*

2. *Sample-Complexity Bound for Importance-Weighted Entropy Estimation*

$$\left|\mathrm{H}\big(p(\boldsymbol{y})\big) - \hat{\mathrm{H}}\big(p(\boldsymbol{y})\big)\right| \leqslant \epsilon \quad \Longleftarrow \quad N \geq \frac{(b-a)^2 C^2}{2\epsilon^2} \log\left(\frac{2}{\delta}\right), \tag{11}$$

   *where $p(\boldsymbol{y})/q(\boldsymbol{y}) \leqslant C$, $\forall \boldsymbol{y} \in \mathcal{Y}_T$.*

*Proof.* 1: Probability of failure for $N$ samples $(1 - Q_\epsilon)^N$, and $\log(1/(1-x)) \leqslant x^{-1}$ for $x \in (0,1)$. 2: Applying Hoeffding's inequality after bounding importance weights by their worst-case value $C$. Detailed derivations are provided in Apx. B. $\qquad\square$

**Discussion.** The sample-complexity bound on estimating $\mathrm{M}(p(\boldsymbol{y}))$ depends on the concentration of output sequences in the $\epsilon$-region. This is desirable for autoregressive language generation, as sampling strategies generally focus on obtaining likely output sequences. In contrast, the sample complexity bound on estimating $\mathrm{H}(p(\boldsymbol{y}))$ depends on the range of possible sequence likelihoods and the worst-case importance weight $C$ squared. Both can be very high in practical settings. Thus, regarding sample-complexity, estimating $\mathrm{M}(p(\boldsymbol{y}))$ appears desirable compared to estimating $\mathrm{H}(p(\boldsymbol{y}))$ for typical LLM output distributions.

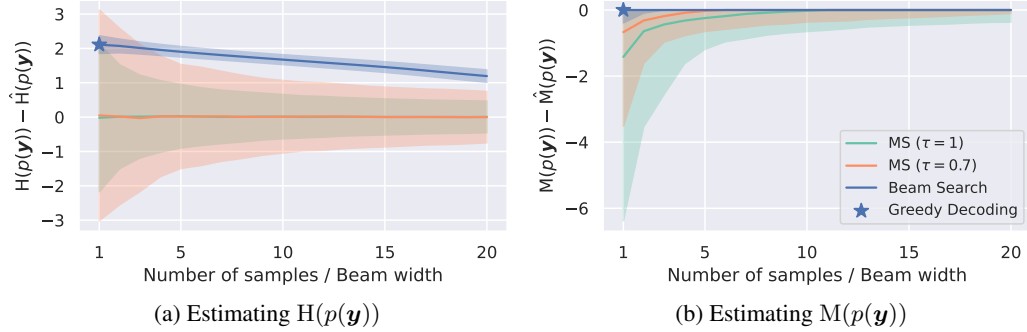

Figure 1: Quality of estimators for synthetic predictive distributions $p(\boldsymbol{y})$ with $|\mathcal{V}| = 20$ and $T = 4$. The predictive entropy $\mathrm{H}(p(\boldsymbol{y}))$ is estimated as in Eq. (4) using multinomial sampling (MS) with different temperatures ($\tau$). The negative maximum sequence log-likelihood $\mathrm{M}(p(\boldsymbol{y}))$ is estimated by the maximum over $N$ samples obtained by beam search ($N = 1$ represents greedy decoding) or MS with different $\tau$. Statistics are obtained by sampling 2000 different $p(\boldsymbol{y})$. (a) Lines show average, shades denote one std. (b) Lines show the median, and shades denote the 5% to 95% quantile range.

**Simulation Study.** To gain insights into the practical implications of Theorem 1, we conduct a synthetic experiment where we sample predictive distributions $p(\boldsymbol{y})$ with smaller vocabulary sizes and shorter sequence lengths, while preserving the distributional characteristics typical for LLMs. This experiment design allows us to obtain ground truths for both $\mathrm{H}(p(\boldsymbol{y}))$ and $\mathrm{M}(p(\boldsymbol{y}))$, which become intractable to compute with larger vocabulary sizes and sequence lengths. Using this synthetic setup, we evaluate how the quality of estimators improves with the number of samples.

Fig. 1a summarizes the results for estimating $\mathrm{H}(p(\boldsymbol{y}))$, derived from the logarithmic score. The results show that low sample sizes lead to high estimator variance. Similarly, Fig. 1b summarizes the results for estimating $\mathrm{M}(p(\boldsymbol{y}))$, derived from the zero-one score. The results indicate that heuristics such as beam search and even greedy decoding provide accurate estimates of $\mathrm{M}(p(\boldsymbol{y}))$ with high probability. In contrast, the variance of estimating $\mathrm{H}(p(\boldsymbol{y}))$ remains substantial even when considering multiple samples and further increases when sampling with a lower temperature. Details on the sampling procedure of the predictive distributions, as well as additional experiments are provided in Apx. C.

## 4 RELATED WORK

In Sec. 1, we discussed prior work that considers single-sequence measures as heuristic baselines (Fadeeva et al., 2023; 2024; Vazhentsev et al., 2024; Abbasi-Yadkori et al., 2024; Plaut et al., 2024; Vashurin et al., 2025a;b), yet there is no clear consensus on how to apply them in a principled way. For instance, Ren et al. (2023) investigate length-normalized sequence likelihood (i.e., perplexity) and Zhang et al. (2025) use likelihood ratios between pre-trained and fine-tuned LLMs for OOD detection, but neither considers the MSP. Likewise, (Chen et al., 2024; Bakman et al., 2024; Yaldiz et al., 2025) include only perplexity as a baseline without specifying how the output sequence is obtained. Qiu and Miikkulainen (2024) also compare against a length-normalized variant, using the normalization from Murray and Chiang (2018), which corrects length bias for neural machine translation rather than uncertainty quantification. By grounding the MSP in theory and guiding its best approximation, we provide a foundation for advancing single-sequence uncertainty measures.

In Sec. 2.2, we also elaborated on the established sampling-based uncertainty estimation measures, i.e., PE (Malinin and Gales, 2018) and SE (Kuhn et al., 2023; Farquhar et al., 2024). There is a body of work that builds upon the concept of PE, for instance, by considering a weighting factor for individual token and sequence likelihoods to account for the importance on a semantic level (Duan et al., 2024; Bakman et al., 2024; Yaldiz et al., 2025). There is also a body of work that extends the concept of SE (Kuhn et al., 2023; Farquhar et al., 2024), for instance, by improving the semantic clustering (Nikitin et al., 2024; Qiu and Miikkulainen, 2024), improving the sampling of diverse output sequences (Aichberger et al., 2025), or directly approximating the measure from hidden states of the language model (Kossen et al., 2024; Chen et al., 2024). Since our goal is to compare the uncertainty principles induced by the underlying proper scoring rules, we focus on their established forms for a fair and consistent comparison.

There is also work on uncertainty estimation in NLG that cannot be directly grounded in proper scoring rules. For instance, several approaches leverage the language model itself to directly predict uncertainty, either through numerical estimates or verbal explanations (Mielke et al., 2022; Lin et al., 2022; Kadavath et al., 2022; Cohen et al., 2023a; Ganguli et al., 2023; Ren et al., 2023; Tian et al., 2023). Cohen et al. (2023b) use cross-examination, where one model generates an output, and another evaluates it, while Manakul et al. (2023) assess uncertainty by feeding multiple sampled outputs into a second model. Zhou et al. (2023) explore the behavior of LLMs when expressing their uncertainty, providing insights into how models articulate confidence. Xiao et al. (2022) analyze how factors such as model architecture and training data affect uncertainty estimates. Gao et al. (2024) perturb inputs to quantify uncertainty, and Chen and Mueller (2024) rely on consistency and self-reflection under different prompting strategies. Jiang et al. (2024) propose Graph Uncertainty, viewing individual claims in an output sequence as nodes in a bipartite graph. Finally, conformal prediction (Quach et al., 2024) provides a calibration-based rule for determining when to stop sampling during generation.

## 5 EXPERIMENTS

We aligned our experiments on evaluating uncertainty estimation methods with prior work by focusing on free-form question answering tasks (Kuhn et al., 2023; Duan et al., 2024; Bakman et al., 2024; Nikitin et al., 2024; Aichberger et al., 2025; Kossen et al., 2024). While Farquhar et al. (2024) additionally concerns experiments with paragraph-length generations, their approach breaks down the paragraph into factual claims and constructs corresponding questions. Therefore, performance on this task is expected to align with general free-form question answering tasks, and we therefore focused on those for a clearer and more direct evaluation. This focus further avoids potential confounding factors introduced by additional experimental complexities.

**Datasets.** We evaluated uncertainty estimation methods on three different datasets. We used the more than 3,000 test instances from *TriviaQA* (Joshi et al., 2017) concerning trivia questions, the more than 300 test instances from *SVAMP* (Patel et al., 2021) concerning elementary-level math problems, and the more than 3,600 test instances from *NQ-Open* (Lee et al., 2019) to assess natural questions aggregated from Google Search. Each dataset was used for two distinct tasks: (1) generating concise answers in the form of short phrases (*short*) and (2) generating more detailed answers in the form of full sentences (*long*), following the experimental setup in Farquhar et al. (2024). The six individual tasks were selected so that they align with prior work to provide a fair and meaningful comparison. They aim at covering a broad range of real-world scenarios, ensuring a comprehensive evaluation.

**Models.** We conducted our evaluations on six different LLMs covering various architectures, sizes, and training stages. Specifically, we used the transformer model series *Llama-3.1* (Dubey et al., 2024) and the state-space model series *Falcon Mamba* (Gu and Dao, 2024; Zuo et al., 2024), representing two prominent paradigms. To assess the effect of training stage and scale on uncertainty estimation in NLG, we considered pre-trained (*PT*) and instruction-tuned (*IT*) LLMs with 7, 8, and 70 billion parameters, covering a wide spectrum of model characteristics used in real-world applications.

**Baselines.** We compare our method, G-NLL, against the commonly used uncertainty measures based on the logarithmic score as of Eq. (4) and Eq. (5) and their variants. These include predictive entropy (*PE*), length-normalized predictive entropy (*LN-PE*) (Malinin and Gales, 2021), semantic entropy (*SE*), length-normalized semantic entropy (*LN-SE*), and Discrete semantic entropy (*D-SE*) (Kuhn et al., 2023; Farquhar et al., 2024). For a given output sequence $\boldsymbol{y}'$, the length-normalized variants consider $\bar{p}(\boldsymbol{y}' \mid \boldsymbol{x}, \boldsymbol{w})$ instead of $p(\boldsymbol{y}' \mid \boldsymbol{x}, \boldsymbol{w})$ to compute the uncertainty estimates. D-SE completely disregards the likelihood of the output sequence and only considers the proportion of output sequences that belong to the same semantic cluster (Farquhar et al., 2024). These baselines represent direct counterparts derived from alternative proper scoring rules as discussed in Sec. 2.2. More importantly, they are among the most widely used uncertainty estimation methods in NLG. Comparing against these established methods provides a decisive proxy for the empirical performance of G-NLL.

**Evaluation.** Effective uncertainty measures should accurately reflect the reliability of answers generated by the language model. In other words, higher uncertainty should correspond to a greater tendency for incorrect outputs. Thus, to evaluate the performance of an uncertainty estimator, we assess how well the uncertainty estimate correlates with the correctness of the language model's answer. Correct answers should be assigned a lower uncertainty estimator than incorrect answers. We consider a given answer correct if the F1 score of the commonly used SQuAD metric to the ground

Table 1: Average AUROC (↑) with standard errors across TriviaQA, SVAMP, and NQ datasets, using uncertainty estimates to distinguish between correct and incorrect answers. Six different language models, with varying model architectures (*transformer*, *state-space*), model sizes (*7B*, *8B*, *70B*), and training stages (*PT*, *IT*) are considered. The reference answer is generated using greedy decoding, either as a whole sentence (*long*) or a short phrase (*short*). Its correctness is assessed by *F1* score using SQuAD > 0.5 as decision rule or LLM-as-a-judge (*LLM*). *PE*, *LN-PE*, *SE*, *LN-SE*, and *D-SE* use 10 output sequences (generated via multinomial sampling) to obtain their uncertainty estimates. G-NLL solely uses one output sequence (generated via greedy decoding) for its uncertainty estimate. * indicates that G-NLL performs significantly better than the best sampling-based method ($p < 0.05$).

| *Underlying proper scoring rule* | | | | Logarithmic | | | | | Zero-One |
|---|---|---|---|---|---|---|---|---|---|
| **Language Model** | | **Gen.** | **Metric** | **PE** | **LN-PE** | **SE** | **LN-SE** | **D-SE** | **G-NLL** |
| **Transformer** 8B | PT | short | F1 | .776 ±.009 | .795 ±.008 | .775 ±.008 | .793 ±.008 | .804 ±.009 | **.843** ±.006 ... |
| | | short | F1 | .776 ±.009 | .795 ±.008 | .775 ±.008 | .793 ±.008 | .804 ±.009 | **.824** ±.008* |
| | | short | LLM | .698 ±.014 | .714 ±.014 | .690 ±.014 | .706 ±.014 | .719 ±.014 | **.726** ±.014 |
| | | long | LLM | .562 ±.012 | .555 ±.012 | .545 ±.013 | .553 ±.012 | .600 ±.012 | **.649** ±.012* |
| | IT | short | F1 | .772 ±.008 | .801 ±.007 | .805 ±.007 | .814 ±.007 | .806 ±.008 | **.838** ±.006* |
| | | short | LLM | .676 ±.015 | .697 ±.015 | .704 ±.015 | .709 ±.015 | .694 ±.016 | **.722** ±.014 |
| | | long | LLM | .551 ±.012 | .548 ±.012 | .599 ±.012 | .601 ±.012 | .609 ±.012 | **.615** ±.012 |
| **Transformer** 70B | PT | short | F1 | .775 ±.010 | .790 ±.010 | .793 ±.009 | .803 ±.009 | .791 ±.009 | **.820** ±.008 |
| | | short | LLM | .693 ±.019 | .709 ±.019 | .718 ±.019 | .722 ±.020 | .715 ±.018 | **.723** ±.019 |
| | | long | LLM | .552 ±.014 | .534 ±.014 | .558 ±.015 | .569 ±.013 | .571 ±.013 | **.649** ±.012* |
| | IT | short | F1 | .748 ±.010 | .781 ±.009 | .790 ±.009 | **.799** ±.009 | .783 ±.009 | .792 ±.011 |
| | | short | LLM | .681 ±.021 | .698 ±.022 | .703 ±.022 | **.709** ±.022 | .699 ±.019 | .699 ±.024 |
| | | long | LLM | .555 ±.013 | .557 ±.014 | .568 ±.014 | .595 ±.014 | **.600** ±.014 | .562 ±.014 |
| **State-Space** 7B | PT | short | F1 | .811 ±.007 | .815 ±.007 | .809 ±.008 | .822 ±.008 | .828 ±.006 | **.843** ±.006* |
| | | short | LLM | .705 ±.012 | .711 ±.011 | .701 ±.012 | .711 ±.012 | .716 ±.012 | **.728** ±.011 |
| | | long | LLM | .567 ±.012 | .597 ±.012 | .574 ±.012 | .611 ±.012 | **.624** ±.012 | .612 ±.012 |
| | IT | short | F1 | .793 ±.009 | .814 ±.007 | .797 ±.008 | .816 ±.008 | .829 ±.007 | **.838** ±.006 |
| | | short | LLM | .690 ±.013 | .701 ±.012 | .689 ±.012 | .699 ±.012 | .711 ±.013 | **.719** ±.012 |
| | | long | LLM | .588 ±.012 | .587 ±.012 | .597 ±.012 | .618 ±.012 | **.629** ±.012 | .615 ±.012 |
| **Average** | | | | .677 ±.003 | .689 ±.003 | .690 ±.003 | .703 ±.003 | .707 ±.003 | **.721** ±.003* |

truth answer exceeds 0.5 (Rajpurkar et al., 2016). Since this metric is only applicable for short-phrase generations that align with the ground truth answer, we additionally employ Llama-3.1 with 70 billion parameters (Dubey et al., 2024) as LLM-as-a-judge to assess the correctness of both short-phrase and full-sentence generations. Finally, to measure the correlation between the incorrectness of answers and the respective uncertainty estimates, we use the AUROC. Higher AUROC values indicate better performance of the uncertainty estimator, as it reflects a stronger alignment between the correctness of the language model's answers and their respective uncertainty estimates. This evaluation process follows established protocols for uncertainty estimation in NLG (Kuhn et al., 2023; Farquhar et al., 2024; Duan et al., 2024; Bakman et al., 2024; Nikitin et al., 2024; Kossen et al., 2024).

## 5.1 MAIN RESULTS

Tab. 1 summarizes the performance of the uncertainty measures across the six LLMs, six tasks, and two evaluation metrics. We report the average AUROC across the three datasets, highlighting the best-performing measure in bold. Additionally, the best-performing measure based on the logarithmic score is underlined, unless it also represents the overall best. In 13 out of 18 scenarios, G-NLL outperforms the uncertainty measures and remains competitive in the remaining 5 scenarios. This strong performance is particularly evident in tasks involving the generation of short phrases, suggesting its effectiveness in capturing the essential part of the output sequence that contains the factual answer to a question. This is especially valuable in practical scenarios, where the uncertainty about a specific fact is often more critical than the uncertainty about the entire generated sentence. Overall, our measure significantly outperforms all other measures when considering the average across all scenarios. This demonstrates that G-NLL achieves strong empirical performance despite relying on only a single output sequence. Crucially, G-NLL relies only on the greedily decoded output sequence, yielding a deterministic and hyperparameter-free uncertainty estimate. Detailed evaluation results are provided in Apx. D.

## 5.2 Ablation Study on Approximation Variants

The strong empirical performance of G-NLL on question answering NLG tasks suggests that it effectively captures key aspects of uncertainty, even when using significantly fewer output sequences compared to uncertainty measures based on the logarithmic score. To examine the underlying factors driving this behavior, we analyze the impact of the sampling method on approximating the measure of aleatoric uncertainty $\mathrm{M}(p(\boldsymbol{y} \mid \boldsymbol{x}, \boldsymbol{w}))$ derived from the zero-one score in Eq. (7).

Specifically, we utilize multinomial sampling (MS) with different temperatures $\tau$, beam search (BS) with different beam sizes, and greedy decoding as for G-NLL. For each sampling method, we generate a single output sequence per instance in the TriviaQA dataset. We then use the corresponding NLL as the uncertainty estimate, following the same evaluation process as in the main experiments above.

Notably, the baselines are unaffected by the choice of sampling method for computing the NLL, as we again use their optimal hyperparameter settings for sampling.

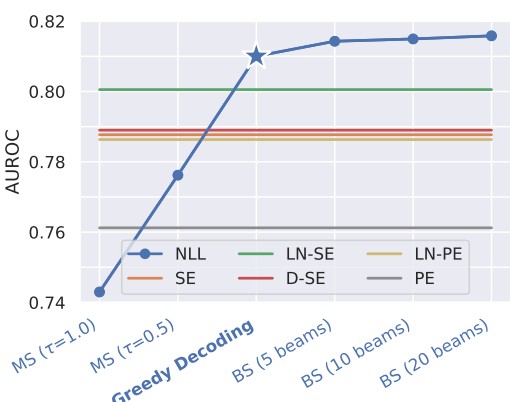

The results in Fig. 2 show that better approximations of the most likely output sequences indeed lead to higher uncertainty estimation performance, reinforcing the validity of our alternative measure derived from the zero-one score. Additionally, it can be observed that sampling output sequences using greedy decoding significantly outperforms MS. While performance improves further with BS, as anticipated, the marginal benefits are relatively small. This supports the claim that greedy decoding provides a strong approximation to the most likely output sequence. Since BS is computationally more expensive (as its beam size corresponds to the number of sampled output sequences), using greedy decoding in G-NLL achieves the best trade-off between effectiveness and efficiency.

Figure 2: Average AUROC (↑) using uncertainty measures based on the zero-one score, with the output sequence to approximate the MSP being generated with multinomial sampling (MS), greedy decoding (G-NLL), or beam search (BS), compared to baseline measures based on the logarithmic score.

## 6 Conclusion

In this work, we theoretically motivate an alternative uncertainty measure: the NLL of the most likely output sequence under a given language model. The measure is grounded in the general notion of proper scoring rules, with the zero-one score serving as an alternative to the commonly adopted logarithmic score. Unlike multi-sequence uncertainty measures, it can be efficiently approximated with G-NLL from a single greedily decoded sequence, challenging the widespread use of sampling- and clustering-based approaches for uncertainty estimation in NLG. Experiments show that G-NLL outperforms prior methods that entail considerably higher computational costs and algorithmic complexity. Importantly, we find that the choice of decoding strategy to approximate the most likely output sequence is critical, and sampling-based or length-normalized variants can degrade approximation quality.

While G-NLL effectively captures uncertainty, it does not explicitly account for semantics. Exploring extensions that incorporate semantic information could further enhance single-sequence uncertainty estimation, but would likely increase computational cost (see Sec. 2.3), and we leave exploring this trade-off for future work. Moreover, G-NLL requires access to the probabilities of generated tokens, though not the full token distribution. While such information is typically exposed by modern LLM APIs, future work may investigate how to approximate it when unavailable.

While there remain opportunities for refinement, G-NLL provides a principled foundation for efficient uncertainty estimation in NLG. Its simplicity and minimal computational overhead make it a practical baseline for developing new uncertainty measures and support scalable deployment in real-world applications. More broadly, our work shows that single-sequence measures serve as a viable alternative to sampling-based methods, offering a principled way to estimate uncertainty efficiently in LLMs.

## ACKNOWLEDGEMENTS

We acknowledge EuroHPC Joint Undertaking for awarding us access to Karolina at IT4Innovations, Czech Republic. The ELLIS Unit Linz, the LIT AI Lab, the Institute for Machine Learning, are supported by the Federal State Upper Austria. We thank the projects FWF AIRI FG 9-N (10.55776/FG9), AI4GreenHeatingGrids (FFG- 899943), Stars4Waters (HORIZON-CL6-2021-CLIMATE-01-01), FWF Bilateral Artificial Intelligence (10.55776/COE12). We thank NXAI GmbH, Audi AG, Silicon Austria Labs (SAL), Merck Healthcare KGaA, GLS (Univ. Waterloo), TÜV Holding GmbH, Software Competence Center Hagenberg GmbH, dSPACE GmbH, TRUMPF SE + Co. KG.

## ETHICS STATEMENT

This work contributes to advancing the efficiency and reliability of uncertainty estimation in LLMs. Our proposed measure, G-NLL, reduces computational overhead compared to existing approaches, enabling broader adoption of uncertainty estimation techniques in real-world applications. While we recognize societal and ethical risks associated with LLMs, such as misinformation, our contribution seeks to mitigate these risks by supporting more trustworthy deployment through improved uncertainty estimation. Our work does not involve human subjects, sensitive data, or personally identifiable information. All datasets used are publicly available and commonly employed in prior work on NLG.

## REPRODUCIBILITY STATEMENT

We provide detailed descriptions of datasets, experimental setup, and evaluation metrics in Sec. 5 of the main paper and Apx. D. All datasets used are publicly available, and we utilize standard benchmarks to support straightforward replication. All experiments were performed on a single node with 8 NVIDIA A100 Tensor Core GPUs. Running all evaluations required roughly 100 node hours.

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

## USAGE OF LARGE LANGUAGE MODELS (LLMs)

LLMs were used as general-purpose assistive tools during the preparation of this paper. Their use fell into two categories: (i) writing assistance, where they were used to improve the clarity and readability of certain passages through language refinement, and (ii) coding assistance, where they were used for support with code completion and debugging. LLMs were not used for research ideation, experimental design, theoretical development, or result analysis. All substantive contributions, including the conception of ideas, methodology, and experiments, were made by the authors.

## A    DETAILS ON PREDICTIVE UNCERTAINTY IN NLG

As stated in Sec. 2, the total uncertainty of a given language model is defined as

$$\mathrm{E}_{p(\tilde{\boldsymbol{w}}|\mathcal{D})}\big[\mathrm{E}_{p(\boldsymbol{y}'|\boldsymbol{x},\boldsymbol{w})}\big[\mathrm{S}\big(p(\boldsymbol{y}\mid\boldsymbol{x},\tilde{\boldsymbol{w}}),\boldsymbol{y}'\big)\big]\big]\ .$$

This corresponds to the expected score that models from the posterior $p(\tilde{\boldsymbol{w}}\mid\mathcal{D})$ assign to output sequences $\boldsymbol{y}'$ drawn from the predictive distribution of the given model $p(\boldsymbol{y}'\mid\boldsymbol{x},\boldsymbol{w})$. Note again that $\boldsymbol{y}'$ is an independent notational copy of $\boldsymbol{y}$ introduced to make the random variable to integrate within the expectation more explicit.

Adding and subtracting $\mathrm{E}_{p(\boldsymbol{y}'|\boldsymbol{x},\boldsymbol{w})}[\mathrm{S}\left(p(\boldsymbol{y}\mid\boldsymbol{x},\boldsymbol{w}),\boldsymbol{y}'\right)]$ yields the general scoring-rule-based uncertainty decomposition in Eq. (1) of the main paper:

$$\underbrace{\mathrm{E}_{p(\tilde{\boldsymbol{w}}|\mathcal{D})}\big[\mathrm{E}_{p(\boldsymbol{y}'|\boldsymbol{x},\boldsymbol{w})}\big[\mathrm{S}\big(p(\boldsymbol{y}\mid\boldsymbol{x},\tilde{\boldsymbol{w}}),\boldsymbol{y}'\big)\big]\big]}_{\text{expected score}}\ =$$

$$\mathrm{E}_{p(\boldsymbol{y}'|\boldsymbol{x},\boldsymbol{w})}\big[\mathrm{S}\big(p(\boldsymbol{y}\mid\boldsymbol{x},\boldsymbol{w}),\boldsymbol{y}'\big)\big]$$
$$+\ \mathrm{E}_{p(\tilde{\boldsymbol{w}}|\mathcal{D})}\big[\mathrm{E}_{p(\boldsymbol{y}'|\boldsymbol{x},\boldsymbol{w})}\big[\mathrm{S}\big(p(\boldsymbol{y}\mid\boldsymbol{x},\tilde{\boldsymbol{w}}),\boldsymbol{y}'\big)\big]\big]\ -\ \mathrm{E}_{p(\boldsymbol{y}'|\boldsymbol{x},\boldsymbol{w})}\big[\mathrm{S}\big(p(\boldsymbol{y}\mid\boldsymbol{x},\boldsymbol{w}),\boldsymbol{y}'\big)\big]\ =$$

$$\underbrace{\mathrm{E}_{p(\boldsymbol{y}'|\boldsymbol{x},\boldsymbol{w})}\big[\mathrm{S}\big(p(\boldsymbol{y}\mid\boldsymbol{x},\boldsymbol{w}),\boldsymbol{y}'\big)\big]}_{\text{entropy term}}+\underbrace{\mathrm{E}_{p(\tilde{\boldsymbol{w}}|\mathcal{D})}\big[\mathrm{E}_{p(\boldsymbol{y}'|\boldsymbol{x},\boldsymbol{w})}\big[\mathrm{S}\big(p(\boldsymbol{y}\mid\boldsymbol{x},\tilde{\boldsymbol{w}}),\boldsymbol{y}'\big)-\mathrm{S}\big(p(\boldsymbol{y}\mid\boldsymbol{x},\boldsymbol{w}),\boldsymbol{y}'\big)\big]\big]}_{\text{divergence term}}.$$

### A.1    UNCERTAINTY MEASURES BASED ON THE LOGARITHMIC SCORE

To derive the established uncertainty measures in Sec. 2.2, we substitute the logarithmic score

$$\mathrm{S}_{\log}\big(p(\boldsymbol{y}\mid\boldsymbol{x},\cdot),\boldsymbol{y}'\big)=-\log p(\boldsymbol{y}=\boldsymbol{y}'\mid\boldsymbol{x},\cdot) \tag{2}$$

into the general scoring-rule-based uncertainty decomposition (Eq. (1)) to get:

$$\mathrm{E}_{p(\tilde{\boldsymbol{w}}|\mathcal{D})}\big[\mathrm{E}_{p(\boldsymbol{y}'|\boldsymbol{x},\boldsymbol{w})}\big[-\log p(\boldsymbol{y}'\mid\boldsymbol{x},\tilde{\boldsymbol{w}})\big]\big]\ =$$

$$\mathrm{E}_{p(\boldsymbol{y}'|\boldsymbol{x},\boldsymbol{w})}\big[-\log p(\boldsymbol{y}'\mid\boldsymbol{x},\boldsymbol{w})\big]\ +\ \mathrm{E}_{p(\tilde{\boldsymbol{w}}|\mathcal{D})}\big[\mathrm{E}_{p(\boldsymbol{y}'|\boldsymbol{x},\boldsymbol{w})}\big[-\log p(\boldsymbol{y}'\mid\boldsymbol{x},\tilde{\boldsymbol{w}})+\log p(\boldsymbol{y}'\mid\boldsymbol{x},\boldsymbol{w})\big]\big].$$

On the LHS, we note the definition of the cross-entropy

$$\mathrm{E}_{p(\boldsymbol{y}'|\boldsymbol{x},\boldsymbol{w})}\big[-\log p(\boldsymbol{y}'\mid\boldsymbol{x},\tilde{\boldsymbol{w}})\big]\ =\ \mathrm{CE}\big(p(\boldsymbol{y}\mid\boldsymbol{x},\boldsymbol{w})\,;\,p(\boldsymbol{y}\mid\boldsymbol{x},\tilde{\boldsymbol{w}})\big)\ .$$

On the RHS, we note that the first term is the definition of the Shannon entropy

$$\mathrm{E}_{p(\boldsymbol{y}'|\boldsymbol{x},\boldsymbol{w})}\big[-\log p(\boldsymbol{y}'\mid\boldsymbol{x},\boldsymbol{w})\big]\ =\ \mathrm{H}\big(p(\boldsymbol{y}\mid\boldsymbol{x},\boldsymbol{w})\big)\ ,$$

and the second term is the definition of the Kullback-Leibler divergence

$$\mathrm{E}_{p(\boldsymbol{y}'|\boldsymbol{x},\boldsymbol{w})}\big[-\log p(\boldsymbol{y}'\mid\boldsymbol{x},\tilde{\boldsymbol{w}})+\log p(\boldsymbol{y}'\mid\boldsymbol{x},\boldsymbol{w})\big]\ =\ \mathrm{E}_{p(\boldsymbol{y}'|\boldsymbol{x},\boldsymbol{w})}\left[\frac{\log p(\boldsymbol{y}'\mid\boldsymbol{x},\boldsymbol{w})}{\log p(\boldsymbol{y}'\mid\boldsymbol{x},\tilde{\boldsymbol{w}})}\right]$$

$$=\ \mathrm{KL}\big(p(\boldsymbol{y}\mid\boldsymbol{x},\boldsymbol{w})\,\big\|\,p(\boldsymbol{y}\mid\boldsymbol{x},\tilde{\boldsymbol{w}})\big)\ .$$

Using these definitions directly results in Eq. (3) of the main paper:

$$\underbrace{\mathrm{E}_{p(\tilde{\boldsymbol{w}}|\mathcal{D})}\big[\mathrm{CE}\big(p(\boldsymbol{y}\mid\boldsymbol{x},\boldsymbol{w})\,;\,p(\boldsymbol{y}\mid\boldsymbol{x},\tilde{\boldsymbol{w}})\big)\big]}_{\text{total uncertainty}}\ = \tag{3}$$

$$\underbrace{\mathrm{H}\big(p(\boldsymbol{y}\mid\boldsymbol{x},\boldsymbol{w})\big)}_{\text{aleatoric uncertainty}}+\underbrace{\mathrm{E}_{p(\tilde{\boldsymbol{w}}|\mathcal{D})}\big[\mathrm{KL}\big(p(\boldsymbol{y}\mid\boldsymbol{x},\boldsymbol{w})\,\big\|\,p(\boldsymbol{y}\mid\boldsymbol{x},\tilde{\boldsymbol{w}})\big)\big]}_{\text{epistemic uncertainty}}\ .$$

## A.2 UNCERTAINTY MEASURES BASED ON THE ZERO-ONE SCORE

To derive the new uncertainty measures in Sec. 2.3, we substitute the zero-one score

$$\mathrm{S}_{\text{0-1}}\big(p(\boldsymbol{y}\mid\boldsymbol{x},\cdot),\boldsymbol{y}'\big) \;=\; 1 \;-\; \mathbb{1}\big\{\boldsymbol{y}' = \underset{\boldsymbol{y}}{\arg\max}\, p(\boldsymbol{y}\mid\boldsymbol{x},\cdot)\big\} \tag{6}$$

into the general scoring-rule-based uncertainty decomposition (Eq. (1)).

For clarity, we distinguish between the most likely output sequence under the given model $\boldsymbol{w}$

$$\boldsymbol{y}^* := \underset{\boldsymbol{y}}{\arg\max}\, p(\boldsymbol{y}\mid\boldsymbol{x},\boldsymbol{w})\,,$$

and the most likely sequence under a model $\tilde{\boldsymbol{w}}$ drawn from the posterior $p(\tilde{\boldsymbol{w}}\mid\mathcal{D})$

$$\tilde{\boldsymbol{y}}^* := \underset{\boldsymbol{y}}{\arg\max}\, p(\boldsymbol{y}\mid\boldsymbol{x},\tilde{\boldsymbol{w}})\,.$$

Because the indicator enforces $\boldsymbol{y}' = \boldsymbol{y}^*$ or $\boldsymbol{y}' = \tilde{\boldsymbol{y}}^*$, the inner expectation over $\boldsymbol{y}'$ collapses, yielding

$$\mathrm{E}_{p(\boldsymbol{y}'\mid\boldsymbol{x},\boldsymbol{w})}\big[\mathrm{S}_{\text{0-1}}\big(p(\boldsymbol{y}\mid\boldsymbol{x},\boldsymbol{w}),\boldsymbol{y}'\big)\big] = 1 - p(\boldsymbol{y}=\boldsymbol{y}^*\mid\boldsymbol{x},\boldsymbol{w})\,,$$
$$\mathrm{E}_{p(\boldsymbol{y}'\mid\boldsymbol{x},\boldsymbol{w})}\big[\mathrm{S}_{\text{0-1}}\big(p(\boldsymbol{y}\mid\boldsymbol{x},\tilde{\boldsymbol{w}}),\boldsymbol{y}'\big)\big] = 1 - p(\boldsymbol{y}=\tilde{\boldsymbol{y}}^*\mid\boldsymbol{x},\boldsymbol{w})\,.$$

Using this these definitions directly results in Eq. (7) of the main paper:

$$\underbrace{\mathrm{E}_{p(\tilde{\boldsymbol{w}}\mid\mathcal{D})}\big[1 - p(\boldsymbol{y}=\tilde{\boldsymbol{y}}^*\mid\boldsymbol{x},\boldsymbol{w})\big]}_{\text{total uncertainty}} = \tag{7}$$

$$\underbrace{1 - p(\boldsymbol{y}=\boldsymbol{y}^*\mid\boldsymbol{x},\boldsymbol{w})}_{\text{aleatoric uncertainty}} + \underbrace{p(\boldsymbol{y}=\boldsymbol{y}^*\mid\boldsymbol{x},\boldsymbol{w}) - \mathrm{E}_{p(\tilde{\boldsymbol{w}}\mid\mathcal{D})}\big[p(\boldsymbol{y}=\tilde{\boldsymbol{y}}^*\mid\boldsymbol{x},\boldsymbol{w})\big]}_{\text{epistemic uncertainty}}\,.$$

**Incorporating Semantics.** As discussed in Sec. 2.1, uncertainty measures are determined by a predictive distribution together with a proper scoring rule. In place of the output-sequence distribution $p(\boldsymbol{y}\mid\boldsymbol{x},\boldsymbol{w})$, we may therefore consider the semantic cluster distribution $p(c\mid\boldsymbol{x},\boldsymbol{w})$ to incorporate semantic information into the uncertainty measures based on the zero-one score.

In this setting, the zero-one score evaluates the predictive distribution at its *most likely semantic cluster* rather than at the most likely output sequence:

$$\mathrm{S}_{\text{0-1}}\big(p(c\mid\boldsymbol{x},\cdot),c'\big) \;=\; 1 \;-\; \mathbb{1}\big\{c' = \underset{c}{\arg\max}\, p(c\mid\boldsymbol{x},\cdot)\big\}\,. \tag{12}$$

Again, $c'$ is an independent notational copy of $c$ introduced to make clear which variable the expectation is calculated for. For clarity, we again distinguish between the most likely semantic cluster under the given model and that under a model drawn from the posterior:

$$c^* := \underset{c}{\arg\max}\, p(c\mid\boldsymbol{x},\boldsymbol{w})\,, \quad \tilde{c}^* := \underset{c}{\arg\max}\, p(c\mid\boldsymbol{x},\tilde{\boldsymbol{w}})\,.$$

Substituting the zero-one score defined in Eq. (12) into the general scoring-rule-based uncertainty decomposition (Eq. (1)) yields the following decomposition of total semantic uncertainty, the expected confidence that the given model assigns to the most likely *semantic cluster* predicted by models drawn from the posterior $p(\tilde{\boldsymbol{w}}\mid\mathcal{D})$:

$$\underbrace{\mathrm{E}_{p(\tilde{\boldsymbol{w}}\mid\mathcal{D})}\big[1 - p(c=\tilde{c}^*\mid\boldsymbol{x},\boldsymbol{w})\big]}_{\text{total semantic uncertainty}} = \tag{13}$$

$$\underbrace{1 - p(c=c^*\mid\boldsymbol{x},\boldsymbol{w})}_{\text{aleatoric semantic uncertainty}} + \underbrace{p(c=c^*\mid\boldsymbol{x},\boldsymbol{w}) - \mathrm{E}_{p(\tilde{\boldsymbol{w}}\mid\mathcal{D})}\big[p(c=\tilde{c}^*\mid\boldsymbol{x},\boldsymbol{w})\big]}_{\text{epistemic semantic uncertainty}}\,.$$

As in Eq. (7), the epistemic semantic uncertainty is a posterior expectation that is challenging to estimate. The aleatoric semantic uncertainty considers the likelihood of the most likely *semantic cluster* under the given language model, i.e, the most-likely cluster probability (MCP), analogous to maximum sequence probability (MSP).

However, approximating the most likely semantic cluster is more challenging, since the semantic cluster distribution $p(c\mid\boldsymbol{x},\boldsymbol{w})$ cannot be sampled from directly (Aichberger et al., 2025). Whether MCP admits an efficient approximation remains an open question. While MSP (as approximated by G-NLL) may correlate with MCP, a detailed analysis of this relationship is left to future work.

# B  DETAILS ON THEORETICAL ANALYSIS

We want to compare the sample complexity of approximating the maximum and expected log-likelihood, showing that approximating the maximum is desirable in the setting of LLMs. In the following, we derive probabilistic concentration bounds on the number of samples $N$ needed to approximate each quantity to within $\epsilon$-precision with high probability and use these to reason about the sample complexity required for reliable approximation.

**Setup.**  As detailed in Sec. 2 in the main paper, we consider the probability distribution $p(\boldsymbol{y} \mid \boldsymbol{x}, \boldsymbol{w})$ of output sequences $\boldsymbol{y}$ induced by the LLM with parameters $\boldsymbol{w}$ for an input sequence $\boldsymbol{x}$. We abbreviate this distribution by $p(\boldsymbol{y})$. We want to study the approximation error for the negative maximum log-likelihood $\mathrm{M}(p(\boldsymbol{y})) = -\max_{\boldsymbol{y}} \log p(\boldsymbol{y})$ (min-entropy) and the negative expected log-likelihood $\mathrm{H}(p(\boldsymbol{y})) = -\mathrm{E}_{p(\boldsymbol{y})}[\log p(\boldsymbol{y})]$ (Shannon entropy) using $N$ samples.

We assume boundedness, i.e. there exist constants $a$, $b$ such that $a \leqslant \log p(\boldsymbol{y}) \leqslant b$ , $\forall \boldsymbol{y} \in \mathcal{Y}_T$. This assumption generally holds in practice as softmax logits are typically clipped and the softmax temperature is non-zero, though the constants can be of large magnitude depending on the sequence length $T$. Furthermore, to incorporate importance sampling schemes (e.g., low-temperature sampling, top-k or top-p sampling, beam search, etc.), we consider access to a proposal distribution $q(\boldsymbol{y})$. This helps us understand how practical decoding choices impact concentration behavior.

## B.1  APPROXIMATING THE MAXIMUM LOG-LIKELIHOOD (MIN-ENTROPY)

Let $\boldsymbol{y}^* = \mathrm{argmax}_{\boldsymbol{y}} p(\boldsymbol{y})$ and $P_\epsilon = \sum_{\boldsymbol{y} \in \mathcal{Y}_\epsilon} p(\boldsymbol{y})$ as well as $Q_\epsilon = \sum_{\boldsymbol{y} \in \mathcal{Y}_\epsilon} q(\boldsymbol{y})$ where $\mathcal{Y}_\epsilon = \{\boldsymbol{y} \in \mathcal{Y}_T \mid \log p(\boldsymbol{y}^*) - \log p(\boldsymbol{y}) \leqslant \epsilon\}$. Thus $\mathcal{Y}_\epsilon$ is the set of output sequences with a log-likelihood within $\epsilon$ of the maximum log-probability, and $P_\epsilon$, $Q_\epsilon$ are the cumulative probability masses under sampling distributions $p$ and $q$ of those output sequences. The correction factor $C_\epsilon = P_\epsilon/Q_\epsilon$ compares the $\epsilon$-regions under distributions $p$ and $q$. If they match, $C_\epsilon \approx 1$, if $q$ is less likely to provide samples within the $\epsilon$-region, $C_\epsilon > 1$ and if it is more likely $C_\epsilon < 1$. We empirically estimate $\mathrm{M}(p(\boldsymbol{y}))$ with $\hat{\mathrm{M}}(p(\boldsymbol{y})) = -\max\{\log p(\boldsymbol{y}^n)\}_{n=1}^N$, the maximum over the log-probabilities of the sampled output sequences.

When sampling from $q(\boldsymbol{y})$, the probability that no $\boldsymbol{y} \in \mathcal{Y}_\epsilon$ is not obtained in $N$ samples is

$$(1 - Q_\epsilon)^N \leqslant \delta . \tag{14}$$

Solving for $N$, we obtain

$$N \geq \frac{\log(1/\delta)}{\log(1/(1 - Q_\epsilon))} . \tag{15}$$

Furthermore, we use $Q_\epsilon = P_\epsilon/C_\epsilon$ and the inequality $\log(1/(1-x)) \leqslant x^{-1}$ for $x \in (0, 1)$ to simplify further.

**Result.**  To ensure with probability at least $1 - \delta$ that $\left|\mathrm{M}(p(\boldsymbol{y})) - \hat{\mathrm{M}}(p(\boldsymbol{y}))\right| \leqslant \epsilon$ it suffices that

$$N \geq \frac{C_\epsilon}{P_\epsilon} \log\left(\frac{1}{\delta}\right) . \tag{16}$$

We will discuss the practical implications of this bound in Sec. B.3.

## B.2  APPROXIMATING THE EXPECTED LOG-LIKELIHOOD (SHANNON ENTROPY)

We consider importance sampling with $q(\boldsymbol{y})$, thus

$$\mathrm{H}(p(\boldsymbol{y})) = -\mathrm{E}_{p(\boldsymbol{y})}[\log p(\boldsymbol{y})] = \mathrm{E}_{q(\boldsymbol{y})}\left[\frac{p(\boldsymbol{y})}{q(\boldsymbol{y})} \log p(\boldsymbol{y})\right]$$

and its MC estimator

$$\hat{\mathrm{H}}(p(\boldsymbol{y})) = -\frac{1}{N} \sum_{n=1}^N \frac{p(\boldsymbol{y}^n)}{q(\boldsymbol{y}^n)} \log p(\boldsymbol{y}^n)$$

with $\boldsymbol{y}^n \sim q(\boldsymbol{y})$. Furthermore, let $c(\boldsymbol{y}) = \frac{p(\boldsymbol{y})}{q(\boldsymbol{y})}$ be bounded, i.e. $c(\boldsymbol{y}) \leqslant C, \forall \boldsymbol{y} \in \mathcal{Y}_T$.

Hoeffding's inequality states that

$$\mathrm{P}\left(\left|\frac{1}{N}\sum_{n=1}^{N}X_n - \mathrm{E}\left[X_n\right]\right| \geq \epsilon\right) \leqslant 2\exp\left(-\frac{2N\epsilon^2}{(b-a)^2}\right), \tag{17}$$

for a random variable $X_n$ with $a \leqslant X_n \leqslant b$. We set $X_n = c(\boldsymbol{y}^n)\log p(\boldsymbol{y}^n)$ such that $aC \leqslant X_n \leqslant bC$ and use the definitions of $\mathrm{H}(p(\boldsymbol{y}))$ and $\hat{\mathrm{H}}(p(\boldsymbol{y}))$, obtaining

$$\mathrm{P}\left(\left|\hat{\mathrm{H}}(p(\boldsymbol{y})) - \mathrm{H}(p(\boldsymbol{y}))\right| \geq \epsilon\right) \leqslant 2\exp\left(-\frac{2N\epsilon^2}{(b-a)^2C^2}\right). \tag{18}$$

We set the r.h.s. $\leqslant \delta$ and solve for $N$ to arrive at the desired bound.

**Result.** To ensure with probability at least $1 - \delta$ that $\left|\hat{\mathrm{H}}(p(\boldsymbol{y})) - \mathrm{H}(p(\boldsymbol{y}))\right| \leqslant \epsilon$, it suffices that

$$N \geq \frac{(b-a)^2C^2}{2\epsilon^2}\log\left(\frac{2}{\delta}\right) \tag{19}$$

Next, we discuss the practical implications of this bound and contrast it to Eq. (16).

### B.3 COMPARING THEORETICAL APPROXIMATION QUALITY

The bounds in Eq. (16) and Eq. (19) quantify the number of samples needed to achieve a given approximation error $\epsilon$ with high probability $1 - \delta$. By analyzing how the bounds scale with properties of $p$ and the sampling distribution $q$, we can compare the relative difficulty of estimating each quantity.

The bound for $\hat{\mathrm{M}}(p(\boldsymbol{y}))$ in Eq. (16) depends only on the concentration of the output sequence distribution along a few probable sequences within the $\epsilon$-region. This is amenable to decoding practices in LLMs such as top-k, top-p, low-temperature, or even beam search, that focus on obtaining very likely output sequences.

The bound for $\hat{\mathrm{H}}(p(\boldsymbol{y}))$ in Eq. (19), in contrast, reflects the variance of the entire distribution and depends on the squared range of $\log p(\boldsymbol{y})$ scaled by the worst-case importance weight $C^2$, both of which can be very high in practice. As it aggregates over the entire support of $p(\boldsymbol{y})$, estimation is sensitive to rare but non-negligible sequences that are unlikely to be sampled under practical sampling strategies.

Regarding importance sampling, approximating $\mathrm{M}(p(\boldsymbol{y}))$ is also desirable. Here, it suffices that the proposal distribution $q$ concentrates around the most likely sequences according to $p$ and there are no importance weights in the calculation of $\hat{\mathrm{M}}(p(\boldsymbol{y}))$. Importance sampling only increases the likelihood of sampling close to the maximum, not the calculation of the estimator itself. In contrast, when approximating $\mathrm{H}(p(\boldsymbol{y}))$, it is necessary to re-weight the samples using importance weights $c(\boldsymbol{y})$. Under a less optimal $q$, this can lead to a strong increase in variance.

Overall, from a sample complexity point of view, approximating $\mathrm{M}(p(\boldsymbol{y}))$ is not only more efficient than approximating $\mathrm{H}(p(\boldsymbol{y}))$, but also well aligned with the way LLMs are used in practice.

## C    DETAILS ON SIMULATION STUDY

In this section, we provide detailed insights into the numerical results presented in Sec. 3, especially Fig. 1.

To recall, we empirically investigate the performance of estimators for the predictive entropy $\mathrm{H}(p(\boldsymbol{y}))$ (Eq. (4)) and the negative maximum sequence log-likelihood (min-entropy) $\mathrm{M}(p(\boldsymbol{y}))$ (Eq. (9)) in a controlled setting. Therefore, we consider a synthetic experiment with the following setup. We are given a space of possible outcomes $\mathcal{V}$ with $|\mathcal{V}| = \{20, 100\}$. The task is to predict a sequence $\boldsymbol{y} = (y_1, ... y_T) \in \mathcal{Y}_T$ where $y \in \mathcal{V}$ and $T$ is 2, 3, or 4. Predictive distributions $p(\boldsymbol{y})$ are not represented by a neural network, but are randomly sampled according to a Dirichlet distribution $\mathrm{Dir}(\{\alpha_1, ..., \alpha_{|\mathcal{V}|}\})$. The alpha parameters of the Dirichlet distribution are specified to yield typical predictive distributions as encountered in LLMs that follow a Zipf distribution. For $|\mathcal{V}| = 20$ we have $\alpha_{1,2} = 10$ and $\alpha_{3-20} = 0.2$. For $|\mathcal{V}| = 100$ we have $\alpha_{1,2} = 10$, $\alpha_{3-6} = 1$ and $\alpha_{7-100} = 0.2$. Note that the order of alpha values is randomly shuffled before drawing each predictive distribution. Representative predictive distributions sampled from this Dirichlet distribution are shown in Fig. 3a and Fig. 3b.

The experiments investigate the quality of the estimators depending on the number of samples $\{\boldsymbol{y}_n\}_{n=1}^N$. This is feasible because the ground truth values for both entropy and maximum sequence log-likelihood can be calculated for this small synthetic example through exhaustive enumeration. For the experiments we present here in the appendix, we average over 2,000 runs, meaning that new sets of samples $\{\boldsymbol{y}_n\}_{n=1}^N$ are drawn to calculate the respective estimators. As beam search is deterministic, it does not vary in this experimental setting, compared to Fig. 1b in the main paper, where we investigated the quality of estimators for different $p(\boldsymbol{y})$.

The results for estimating the entropy are shown in Fig. 4. We observe that the variance of estimators increases for larger vocabulary sizes $|\mathcal{V}|$ and sequence lengths $T$. Furthermore, lower temperatures increase the variance of the estimator as expected. Note that we introduced clipping for importance weights at $1e \pm 6$, which did not introduce noticeable bias, but made results numerically stable.

The results for estimating the maximum sequence likelihood are shown in Fig. 5. We observe that low-temperature multinomial sampling and beam search find the maximum sequence log-likelihood with a low number of samples with high probability. Greedy decoding (beam width of 1) finds the maximum for all experimental settings except one, where it takes a beam width of 2 to find it with high probability.

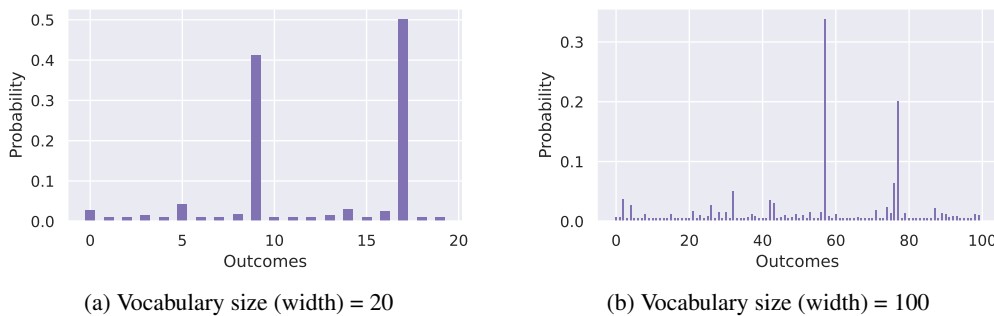

(a) Vocabulary size (width) = 20          (b) Vocabulary size (width) = 100

Figure 3: Exemplary predictive distributions $p(y_t \mid \boldsymbol{y}_{<t})$ for different vocabulary sizes (widths).

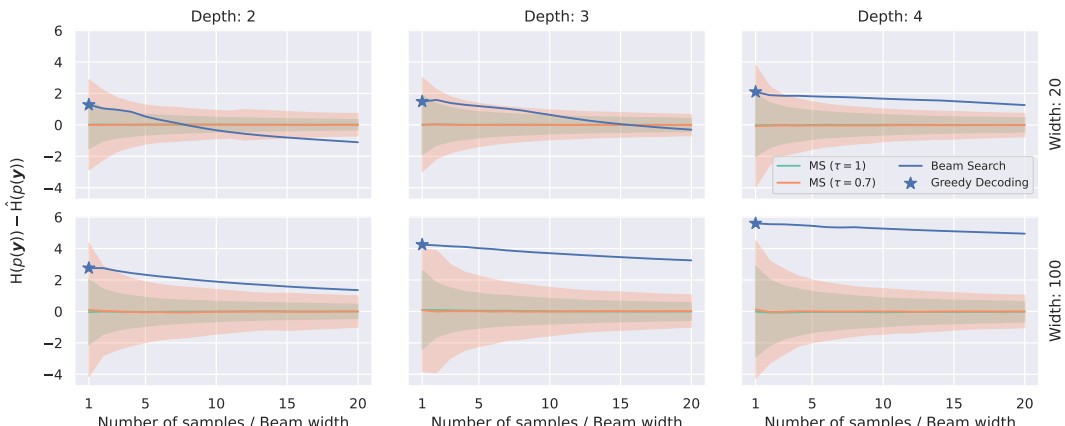

Figure 4: **Estimator of Predictive Entropy.** Results for different vocabulary sizes (width) and sequence lengths (depth). We estimate the entropy $\mathrm{H}(p(\boldsymbol{y}))$ using $N$ Monte-Carlo samples (cf. Eq. (4)). Lines denote the average over runs, while shades denote one standard deviation. We compare MS for two commonly used $\tau$. The experiments show that the decreased temperature leads to lower variance but introduces bias.

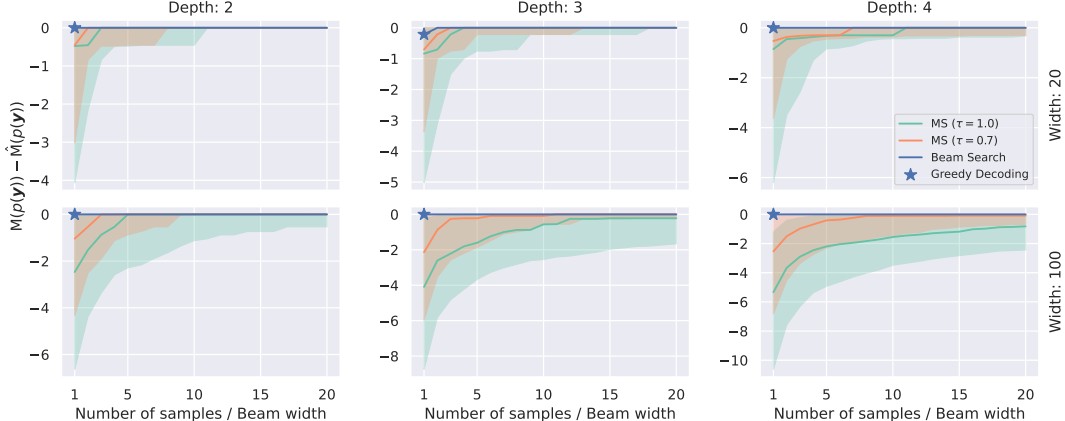

Figure 5: **Estimator of maximum sequence log-likelihood.** Results for different vocabulary sizes (width) and sequence lengths (depth). We estimate $\mathrm{M}(p(\boldsymbol{y}))$ using the maximum over $N$ sampled obtained by beam search ($N = 1$ is greedy decoding) or MS with different $\tau$. Lines denote the median, and shades signify the possible values between the 5 and 95 percent quantiles. Beam search is deterministic for any given distribution $p(\boldsymbol{y})$. Even with a very low number of samples, low-temperature MS and beam search are able to find the maximum with high probability.

# D    DETAILS ON EXPERIMENTAL RESULTS

In this section, we provide detailed insights to complement the main results presented in Sec. 5.1. The code and data are available at `https://github.com/ml-jku/G-NLL`.

**Hyperparameters.**    To compute G-NLL, we use greedy decoding to generate the reference answer, which is equivalent to beam search with a single beam and multinomial sampling with a sampling temperature of zero. Notably, G-NLL is deterministic and hyperparameter-free. To compute the logarithmic score based measures (*PE*, *LN-PE*, *SE*, *LN-SE*, *D-SE*), 10 output sequences are generated via multinomial sampling. For each dataset, we performed a hyperparameter search on held-out instances to determine the best-performing temperature $\tau \in \{0.5, 1.0, 1.5\}$ for sampling output sequences used for the logarithmic score based measures.

**Evaluation Metrics.**    The correctness of the reference answer is assessed by checking if the F1 score of the commonly used SQuAD metric between the reference answer and the given answer exceeds 0.5 (*F1*). Although there are some limitations to using such a simple metric, it has relatively small errors in standard datasets and, therefore, remains widely used in practice. Additionally, we use the model Llama-3.1-70B as LLM-as-a-judge to assess if the given answer is correct (*LLM*).

AUROC is used as the primary performance metric throughout this paper, consistent with standard evaluation practices in this field. We report the results for the individual datasets in Tab. 2, which have been averaged over in Tab. 1. In addition to AUROC, we also consider the average rejection accuracy, i.e., the accuracy of model predictions when allowing the rejection of a certain budget of predictions based on the uncertainty estimate. Results are presented in Tab. 3, where predictions are only evaluated for the 80% most certain predictions, and we again use greedy decoding for our measure based on the zero-one score. This further suggests that our measure remains highly competitive across various settings.

**Evaluation Tasks.**    We evaluate our method on both short phrase answers and full-sentence answers (referred to as "short" and "long" respectively), assessing the generalization of uncertainty estimation across model classes, model sizes, training stages, and evaluation criteria for output correctness, using three different question-answering datasets. We selected these datasets to best align with prior work to provide a fair and meaningful comparison (Duan et al., 2024; Kuhn et al., 2023; Farquhar et al., 2024; Bakman et al., 2024; Nikitin et al., 2024; Aichberger et al., 2025; Kossen et al., 2024). Notably, Farquhar et al. (2024) additionally investigates paragraph-length summarization of biographies by decomposing the task into individual question-answering problems. This suggests that performance on question-answering tasks is expected to be indicative of performance on summarization tasks as well. As such, our experimental setup reflects a wide spectrum of practical tasks.

**Length-Normalization.**    Intuitively, length normalization might appear to be a reasonable choice also for the single-sequence measure. However, we find that G-NLL without length normalization yields the best performance, even on tasks with high variability in sequence length, as reported in Tab. 4. This can be attributed to the fact that length normalization tends to dilute the influence of low-probability tokens. Since most tokens typically have relatively high likelihoods, summing log-probabilities (rather than averaging them) places greater emphasis on the more uncommon, low-likelihood tokens that are more informative for uncertainty estimation. Although not specific to single-sequence measures, investigating the trade-off introduced by length normalization to handle these characteristics represents a promising direction for future work.

**Beam-Search for MSP Approximation.**    We investigate how much the alternative measure derived from the zero-one score benefits from better approximating the most likely output sequences (MSP), as searching for more likely output sequences through beam search theoretically further improves the approximation of MSP. The results summarized in Tab. 5 show that the performance does not significantly improve compared to greedy decoding, which is consistent with the ablation study presented in Sec. 5.2. This further supports the claim that G-NLL is a strong measure of uncertainty, despite its algorithmic simplicity and computational efficiency.

Table 2: **AUROC Evaluation.** Average AUROC (↑) using uncertainty estimates of different measures as a score to distinguish between correct and incorrect answers of each dataset. The reference answer is generated using greedy decoding, either as a whole sentence (*long*) or a short phrase (*short*).

| $\mathcal{D}$ | Language Model | | Gen. | Metric | PE | LN-PE | SE | LN-SE | D-SE | G-NLL |
|---|---|---|---|---|---|---|---|---|---|---|
| *Uncertainty measure generating scoring rule* | | | | | *Logarithmic* | | | | | *Zero-One* |
| TriviaQA | Transformer 8B | PT | short | F1 | .758 | .778 | .788 | .798 | .787 | **.810** |
| | | | short | LLM | .675 | .694 | .703 | .704 | .682 | **.722** |
| | | | long | LLM | .592 | .604 | .640 | .631 | .650 | **.704** |
| | | IT | short | F1 | .735 | .768 | .790 | .800 | .777 | **.809** |
| | | | short | LLM | .660 | .684 | .708 | .710 | .680 | **.716** |
| | | | long | LLM | .603 | .627 | **.678** | .672 | .670 | .670 |
| | Transformer 70B | PT | short | F1 | .707 | .730 | .741 | .743 | .702 | **.744** |
| | | | short | LLM | .650 | .660 | .696 | .695 | .656 | **.698** |
| | | | long | LLM | .538 | .533 | .625 | .574 | .563 | **.692** |
| | | IT | short | F1 | .698 | .714 | .722 | **.726** | .688 | .722 |
| | | | short | LLM | .663 | .675 | .685 | .679 | .633 | **.701** |
| | | | long | LLM | .530 | .553 | .564 | **.571** | .564 | .543 |
| | State-Space 7B | PT | short | F1 | .786 | .793 | .812 | .818 | .810 | **.832** |
| | | | short | LLM | .687 | .697 | .712 | .714 | .695 | **.724** |
| | | | long | LLM | .597 | .653 | .675 | .680 | .689 | **.705** |
| | | PT | short | F1 | .780 | .799 | .810 | .819 | .811 | **.827** |
| | | | short | LLM | .696 | .701 | .714 | .717 | .703 | **.730** |
| | | | long | LLM | .645 | .654 | .688 | **.698** | .692 | .694 |
| SVAMP | Transformer 8B | PT | short | F1 | .847 | .867 | .865 | .870 | .868 | **.885** |
| | | | short | LLM | .779 | .788 | .753 | .772 | **.791** | .772 |
| | | | long | LLM | .575 | .563 | .519 | .534 | .601 | **.669** |
| | | IT | short | F1 | .879 | .903 | .914 | .912 | .887 | **.931** |
| | | | short | LLM | .706 | .725 | .736 | .731 | .701 | **.753** |
| | | | long | LLM | .556 | .524 | .590 | .608 | .631 | **.662** |
| | Transformer 70B | PT | short | F1 | .892 | .906 | .925 | .929 | .923 | **.936** |
| | | | short | LLM | .794 | .817 | .814 | .815 | **.819** | .799 |
| | | | long | LLM | .578 | .554 | .553 | .579 | .571 | **.665** |
| | | IT | short | F1 | .830 | .895 | .915 | **.922** | .915 | .909 |
| | | | short | LLM | .703 | .744 | .734 | .748 | **.762** | .713 |
| | | | long | LLM | .601 | .577 | .613 | .649 | **.663** | .597 |
| | State-Space 7B | PT | short | F1 | .882 | .893 | .874 | .883 | .889 | **.914** |
| | | | short | LLM | .752 | .757 | .730 | .738 | .757 | **.776** |
| | | | long | LLM | .536 | .585 | .534 | .602 | **.612** | .579 |
| | | IT | short | F1 | .843 | .891 | .854 | .876 | .892 | **.905** |
| | | | short | LLM | .706 | .730 | .704 | .709 | .737 | **.744** |
| | | | long | LLM | .577 | .586 | .578 | .616 | **.639** | .613 |
| NQ | Transformer 8B | PT | short | F1 | .725 | .739 | .673 | .710 | .758 | **.776** |
| | | | short | LLM | .639 | .661 | .615 | .641 | **.683** | **.683** |
| | | | long | LLM | .517 | .498 | .478 | .495 | .550 | **.573** |
| | | IT | short | F1 | .702 | .732 | .711 | .731 | .756 | **.774** |
| | | | short | LLM | .662 | .682 | .669 | .685 | **.700** | .697 |
| | | | long | LLM | .494 | .491 | **.530** | .524 | .527 | .514 |
| | Transformer 70B | PT | short | F1 | .727 | .733 | .711 | .737 | .748 | **.779** |
| | | | short | LLM | .634 | .649 | .642 | .657 | .671 | **.672** |
| | | | long | LLM | .538 | .514 | .494 | .553 | .580 | **.589** |
| | | IT | short | F1 | .718 | .734 | .734 | **.748** | .746 | .743 |
| | | | short | LLM | .676 | .674 | .689 | .698 | **.702** | .681 |
| | | | long | LLM | .535 | .540 | .526 | .566 | **.574** | .545 |
| | State-Space 7B | PT | short | F1 | .766 | .758 | .741 | .765 | **.785** | .782 |
| | | | short | LLM | .675 | .680 | .661 | .681 | **.697** | .683 |
| | | | long | LLM | .567 | .553 | .512 | .551 | **.572** | .554 |
| | | IT | short | F1 | .755 | .751 | .727 | .754 | **.783** | .781 |
| | | | short | LLM | .669 | .672 | .648 | .671 | **.692** | .683 |
| | | | long | LLM | .541 | .521 | .526 | .541 | **.554** | .537 |
| **Average** | | | | | .677 | .689 | .690 | .703 | .707 | **.721** |

Table 3: **Rejection Accuracy Evaluation.** Average Rejection Accuracy at 80% (↑) across all datasets, using uncertainty estimates of different measures as a score to distinguish between correct and incorrect answers. The reference answer is generated using greedy decoding, either as a whole sentence (*long*) or a short phrase (*short*).

| *Uncertainty measure generating scoring rule* | | | *Logarithmic* | | | | | *Zero-One* |
|---|---|---|---|---|---|---|---|---|
| **Language Model** | **Gen.** | **Metric** | **PE** | **LN-PE** | **SE** | **LN-SE** | **D-SE** | `G-NLL` |
| **Transformer** **8b** PT | short | F1 | .661 | .672 | .651 | .643 | .655 | **.681** |
| | | LLM | .774 | **.782** | .767 | .766 | .765 | .778 |
| | | LLM-Instruct | .704 | .721 | .693 | .688 | .702 | **.723** |
| | long | LLM | .596 | .590 | .598 | .592 | .590 | **.619** |
| | | LLM-Instruct | .667 | .684 | .632 | .643 | .644 | **.686** |
| **8b** IT | short | F1 | .668 | .684 | .680 | .673 | .687 | **.702** |
| | | LLM | .775 | .781 | .779 | .775 | .778 | **.788** |
| | | LLM-Instruct | .723 | .742 | .732 | .726 | .743 | **.751** |
| | long | LLM | .628 | .630 | .651 | .644 | **.653** | .652 |
| | | LLM-Instruct | .713 | .724 | .705 | .713 | .727 | **.734** |
| **70b** PT | short | F1 | .818 | .827 | .822 | .827 | .829 | **.836** |
| | | LLM | .844 | .852 | .846 | .847 | .851 | **.855** |
| | | LLM-Instruct | .867 | .875 | .876 | .881 | **.885** | .881 |
| | long | LLM | .704 | .699 | .719 | .707 | .705 | **.724** |
| | | LLM-Instruct | .789 | .795 | .776 | .781 | .788 | **.812** |
| **70b** IT | short | F1 | .795 | .813 | .814 | .809 | .819 | **.823** |
| | | LLM | .836 | .842 | .842 | .837 | .844 | **.845** |
| | | LLM-Instruct | .850 | .867 | .866 | .865 | **.874** | .870 |
| | long | LLM | .706 | .706 | .712 | .715 | **.721** | .715 |
| | | LLM-Instruct | .855 | .850 | .827 | .842 | **.861** | .851 |
| **State-Space** **7b** PT | short | F1 | .598 | .596 | .585 | .579 | .583 | **.612** |
| | | LLM | .729 | .737 | .723 | .721 | .733 | **.742** |
| | | LLM-Instruct | .638 | .640 | .626 | .621 | .632 | **.651** |
| | long | LLM | .613 | **.627** | .612 | .624 | .620 | .623 |
| | | LLM-Instruct | .606 | .611 | .601 | .611 | .618 | **.633** |
| **7b** IT | short | F1 | .592 | .603 | .588 | .581 | .589 | **.615** |
| | | LLM | .737 | .742 | .730 | .726 | .740 | **.744** |
| | | LLM-Instruct | .632 | .646 | .625 | .619 | .637 | **.653** |
| | long | LLM | .611 | .617 | .618 | .612 | **.625** | .625 |
| | | LLM-Instruct | .643 | .652 | .628 | .628 | .654 | **.658** |
| **Average** | | | .712 | .720 | .711 | .710 | .718 | **.729** |

Table 4: **Length Normalization Evaluation.** Average AUROC (↑) across TriviaQA, SVAMP, and NQ datasets, utilizing G‑NLL and its length-normalized version LN‑G‑NLL to assign an uncertainty estimate, which is used as a score to distinguish between correct and incorrect answers. The reference answer is generated using greedy decoding, either as a whole sentence (*long*) or a short phrase (*short*).

| *Uncertainty measure generating scoring rule* | | | | *Zero-One* | |
|---|---|---|---|---|---|
| **Language Model** | | **Generation** | **Metric** | **LN‑G‑NLL** | **G‑NLL** |
| Transformer | 8B PT | short | F1 | .811 | **.824** |
| | | short | LLM | .717 | **.726** |
| | | long | LLM | .532 | **.649** |
| | 8B IT | short | F1 | .826 | **.838** |
| | | short | LLM | .716 | **.722** |
| | | long | LLM | .542 | **.615** |
| | 70B PT | short | F1 | .811 | **.820** |
| | | short | LLM | .718 | **.723** |
| | | long | LLM | .529 | **.649** |
| | 70B IT | short | F1 | .788 | **.792** |
| | | short | LLM | .695 | **.699** |
| | | long | LLM | .539 | **.562** |
| State-Space | 7B PT | short | F1 | .828 | **.843** |
| | | short | LLM | .720 | **.728** |
| | | long | LLM | .576 | **.612** |
| | 7B IT | short | F1 | .823 | **.838** |
| | | short | LLM | .713 | **.719** |
| | | long | LLM | .562 | **.615** |

Table 5: **Beam Search Evaluation.** Average AUROC (↑) across all datasets, using uncertainty estimates of different measures as a score to distinguish between correct and incorrect answers. The reference answer is generated using *beam search with 5 beams*, again either as a whole sentence (*long*) or a short phrase (*short*).

| *Uncertainty measure generating scoring rule* | | | | *Logarithmic* | | | | | *Zero-One* |
|---|---|---|---|---|---|---|---|---|---|
| **Language Model** | | **Gen.** | **Metric** | **PE** | **LN-PE** | **SE** | **LN-SE** | **D-SE** | **G‑NLL** |
| Transformer | 8B PT | short | F1 | .775 | .791 | .765 | .787 | .799 | **.822** |
| | | short | LLM | .700 | .712 | .686 | .704 | .713 | **.726** |
| | | long | LLM | .556 | .540 | .493 | .520 | .578 | **.591** |
| | 8B IT | short | F1 | .778 | .808 | .805 | .819 | .811 | **.845** |
| | | short | LLM | .682 | .704 | .706 | .713 | .698 | **.729** |
| | | long | LLM | .535 | .520 | .584 | .585 | **.586** | .559 |
| | 70B PT | short | F1 | .788 | .799 | .796 | .812 | .798 | **.833** |
| | | short | LLM | .700 | .717 | .719 | **.727** | .718 | .725 |
| | | long | LLM | .540 | .552 | .489 | .531 | .552 | **.608** |
| | 70B IT | short | F1 | .756 | .786 | .796 | **.806** | .788 | .800 |
| | | short | LLM | .680 | .697 | .701 | **.707** | .695 | **.707** |
| | | long | LLM | .534 | .533 | .544 | .569 | **.574** | .534 |
| State-Space | 7b PT | short | F1 | .814 | .818 | .806 | .823 | .825 | **.846** |
| | | short | LLM | .703 | .709 | .699 | .711 | .712 | **.719** |
| | | long | LLM | .570 | .595 | .550 | **.609** | .602 | .563 |
| | 7b IT | short | F1 | .799 | .815 | .794 | .817 | .828 | **.845** |
| | | short | LLM | .699 | .713 | .694 | .709 | .720 | **.730** |
| | | long | LLM | .574 | .575 | .582 | **.621** | .607 | .577 |
| **Average** | | | | .677 | .688 | .678 | .698 | .700 | **.709** |

