# OpenReview forum: "Rethinking Uncertainty Estimation in LLMs: A Principled Single-Sequence Measure"
_ICLR.cc/2026/Conference — ICLR 2026 Poster_

### Official Review · Reviewer_rQPZ · 2025-10-31

**Soundness:** 3
**Presentation:** 2
**Contribution:** 2
**Rating:** 4
**Confidence:** 4

**Summary:**

The paper presents a new uncertainty quantification scheme for LLMs that leverages the probability of the most likely sequence generated for the input prompt. The proposed method is formulated in terms of a proper scoring rule, specifically a zero-one score. This kind of framing comes with certain properties. The experimental evaluation is restricted to QA benchmark datasets and 3 open models from the Llama and Falcon families. The results show indeed improved performance over the competitive approaches, some of them requiring multiple samples.

**Strengths:**

- Uncertainty quantification for LLM is currently a hot topic and therefore advances in this area are warranted.

- The proposed method is very efficient in the sense that it requires a single greedy decoding.

**Weaknesses:**

- I found the presentation of sections 2 and 3 not very clear. Perhaps an illustrative example would help to better convey the technical details. Also, it is not clear what is new here, as proper scoring rules have been presented elsewhere.

- The performance improvements appear to be just marginal over the competitive approaches e.g., 0.804 vs 0.824 in Table 3. Standard deviations appear to be missing from the table, and these need to be included to account for the noise. Also, for such mediocre performance bumps, statistical significance tests are mandatory.

- The proposed method assumes access to sequence probabilities and therefore it is not applicable to closed models such as GPT or Gemini. I think this limitation should be made clearer in the presentation.

**Questions:**

- I may have misunderstood, but Eq 10 looks like summing up the log probabilities of the tokens obtained during greedy decoding. If that's the case, then it looks like your proposed approach is a slight reformulation of previous methods that combine token probabilities in different ways.

---

> ### Author Response · Authors · 2025-11-20
>
> We thank you for the time and effort you put into assessing our work and appreciate your positive remarks on the efficiency of G-NLL.
>
> Below, we would like to address the remaining concerns and questions you raised:
>
> ---
>
> > I found the presentation of sections 2 and 3 not very clear. Perhaps an illustrative example would help to better convey the technical details. Also, it is not clear what is new here, as proper scoring rules have been presented elsewhere.
>
> In Sec. 2, our goal was to show how MSP arises as the proper uncertainty measure induced by the zero-one scoring rule, a connection that has not been made in prior work, and to demonstrate how G-NLL provides a tractable approximation.
>
> In Sec. 3, we aimed to give an intuitive proof sketch of the sample complexity bounds, clarify their implications, and support them with synthetic experiments, while keeping the full derivations in the appendix to maintain readability.
>
> To improve clarity, we expanded the explanation of how G-NLL emerges as an approximation to MSP in the updated version of the paper (see lines 250 ff. highlighted in green). If specific steps remain unclear, we would appreciate pointers and are happy to add further clarification or an illustrative example where needed.
>
> ---
>
> > The performance improvements appear to be just marginal over the competitive approaches e.g., 0.804 vs 0.824 in Table 3. Standard deviations appear to be missing from the table, and these need to be included to account for the noise. Also, for such mediocre performance bumps, statistical significance tests are mandatory.
>
> We agree with your point that standard errors and significance tests help further strengthen the evaluation. Thus, we have updated Tab. 3 to report standard errors using the bootstrapped evaluation procedure of Farquhar et al. (2024). Additionally, we conducted statistical significance tests which indicate that G-NLL is significantly better in five settings, and no sampling-based method is significantly better than G-NLL in any setting. Furthermore, when **integrating over datasets and models (36 considered settings), G-NLL remains statistically significantly better than the best sampling-based alternative (p = 0.0005), while requiring only a single forward pass and no hyperparameters to be tuned**.
>
> We would also like to highlight that the **sampling-based methods have roughly 10 times higher computational cost**, which makes G-NLL valuable even if it only matched their performance, and even more so given that it outperforms them.
>
> ---
>
> > The proposed method assumes access to sequence probabilities and therefore it is not applicable to closed models such as GPT or Gemini. I think this limitation should be made clearer in the presentation.
>
> It is true that G-NLL assumes access to log-probabilities. However, unlike many previous methods, it only requires the log-probabilities of the generated tokens rather than access to the full token distribution, which is typically available through modern LLM APIs. We clarified this in the updated version of the paper (lines 530 ff. highlighted in green).
>
> ---
>
> > I may have misunderstood, but Eq 10 looks like summing up the log probabilities of the tokens obtained during greedy decoding. If that's the case, then it looks like your proposed approach is a slight reformulation of previous methods that combine token probabilities in different ways.
>
> It is correct that at the implementation level G-NLL is a summation of log-probabilities of the tokens produced by greedy decoding. Our main contribution, however, is a deeper insight into
> - how the underlying measure can be theoretically grounded (Sec. 2),
> - which type of uncertainty this measure is quantifying (Sec. 2.3),
> - how its sample complexity compares to entropy-based measures (Sec. 3), and
> - how it can be approximated most effectively via G-NLL (Sec. 5.2).
>
> Additionally, we show that G-NLL can even outperform computationally far more expensive uncertainty methods (Tab. 1), which have been published at top venues, such as ICLR and Nature, without considering this simple single-sequence baseline [1, 2]. This further underscores the importance of theoretically grounding this single-sequence measure and discussing the practical value of G-NLL.
>
> ---
>
> We hope our responses and the updated version of the paper address the raised points clearly. If there is anything that would benefit from further clarification, we would be glad to continue the discussion. Otherwise, we appreciate your reconsideration of our work.
>
> ---
>
> [1] L. Kuhn, Y. Gal, and S. Farquhar, "Semantic uncertainty: Linguistic invariances for uncertainty estimation in natural language generation," ICLR, 2023.
>
> [2] S. Farquhar, J. Kossen, L. Kuhn, and Y. Gal, "Detecting hallucinations in large language models using semantic entropy," Nature, 2024.

---

### Official Review · Reviewer_4Jjw · 2025-10-31

**Soundness:** 3
**Presentation:** 3
**Contribution:** 2
**Rating:** 6
**Confidence:** 4

**Summary:**

This paper proposes and justifies a G-NLL measure for uncertainty estimation in LLMs. The main contribution is showing the connection to an AU like term while computing the expected score for the zero-one scoring rule. As I understand, the measure itself has been explored before, but this work provides some justification. Experiments are conducted to show the merits of the simple measure over some baselines. I have reviewed a prior version of the paper and find that this revised version is extremely similar. In fact, I’m not sure about the edits that have been made in this version – they seem minor. Overall, I like aspects of this paper and think it is a useful contribution.

**Strengths:**

The main strength of the paper is its contribution of making a justification for the proposed uncertainty measure G-NLL. It does so by showing the measure to approximate the aleatoric term in the Bayesian decomposition (TU = AU + EU) for the expected score according to a specific scoring rule. Overall, I continue to appreciate the overall simplicity of the final solution.

**Weaknesses:**

I find the additional contribution to practice to be somewhat limited, as it seems that the measure or related measures have been explored in prior work.

Also, I find the claims by the paper to be somewhat excessive – the measure is made out to be more useful than I believe it really is. See my comment about this later.

**Questions:**

Some comments/questions follow:

G-NLL is said to approximate MSP. Could the authors clarify this a bit further or point me to the place in the paper where they explain the relation?

I feel it would help to present a much broader view of UQ; the current scope is heavily biased towards a subset of literature. This is a suggestion that I made previously as well, and I am slightly frustrated that it seems to have been ignored. Please see some recent survey papers for ideas if needed.

It appears to be standard practice for many uncertainty measures in the literature to be used as scores for predicting whether the greedy sample is correct or not, using metrics like AUROC. Could the authors share their thoughts on why this is the standard mode of evaluation? I am curious to hear their perspective.

“Our work challenges the reliance on sampling-based methods and offers a principled way for efficient uncertainty estimation for LLMs.” I consider the first part of this statement to be a stretch. There is a much broader spectrum of sampling-based methods that were not explored. Moreover, only 3 datasets were considered for experiments, and they do not even scratch the surface of breadth of potential applications. Also, the proposed method only works for scoring the greedy sample and not an arbitrary sample, making uncertainty estimation itself somewhat limiting. In short, there are numerous limitations for the scope of the work. I recommend modulating this statement and discussing limitations adequately.

---

> ### Author Response · Authors · 2025-11-20
>
> We thank you for the thorough assessment of our work and appreciate your positive remarks on it being a "*useful contribution*" and on the "*overall simplicity of the final solution*".
>
> In the following, we would like to address the questions you raised:
>
> ---
>
> > G-NLL is said to approximate MSP. Could the authors clarify this a bit further or point me to the place in the paper where they explain the relation?
>
> Let’s recall that the maximum sequence probability (MSP) is propositional to the NLL of the most likely output sequence: $- max_{y} \sum_t log p(y_t | y_{<t}, x, w)$, as shown in Eq.(9).
>
> G-NLL approximates Eq.(9) by selecting the most likely token at each decoding step: $- max_{y} \sum_t log p(y_t | y_{<t}, x, w) \approx - \sum_t max_{y_t} log p(y_t | y_{<t}, x, w)$, as defined in Eq.(10).
>
> Intuitively speaking, since the $\max$ operation in Eq.(9) requires considering the entire space of output sequences $y$, it is intractable to compute. To make the computation tractable, we propose replacing the full sequence maximization with a tokenwise maximization by moving the $\max$ operator inside the summation. This yields a very efficient approximation, since it corresponds to standard greedy decoding with the given language model.
>
> We **expanded Eq. (9) and the discussion prior to Eq. (10)** to make it clearer how G-NLL emerges as an approximation to MSP (lines 252 ff. highlighted in green). In the remainder of the paper, we show that this is a strong approximation.
>
> ---
>
> > I feel it would help to present a much broader view of UQ [...]
>
> Due to space constraints, we had to focus our discussion in the main paper on the most directly relevant methods (see Sec. 1 and Sec. 2.1) and decided to move the broader overview to the appendix (see Sec. A of the original version).
> That said, we now used the additional page allowance of the rebuttal to **added Sec. 4, “Related Work”**, in the updated version of the main paper. We would welcome your feedback and thoughts on any additional streams of works you believe should be discussed.
>
> ---
>
> > It appears to be standard practice for many uncertainty measures in the literature to be used as scores for predicting whether the greedy sample is correct or not, using metrics like AUROC. [...]
>
> Regarding the use of AUROC, it is a standard way to quantify how well an uncertainty score separates correct from incorrect outputs. In general, correct answers should be assigned a lower uncertainty estimator than incorrect ones. Also, if a score is strongly correlated with correctness, it can be used in practice to decide whether to trust a model’s answer.
> As for why the greedy output is typically used as reference, there are two main reasons. First, it removes randomness from the evaluation. Second, this follows the convention from misclassification detection in classification models, where correctness is judged based on the argmax prediction. If any part of this is still unclear or you have a different perspective, we would be glad to discuss it further.
>
> ---
>
> > “Our work challenges the reliance on sampling-based methods and offers a principled way for efficient uncertainty estimation for LLMs.” I consider the first part of this statement to be a stretch.
>
> We would like to clarify that this statement was not intended to claim superiority over all sampling-based methods, but rather to encourage readers to reconsider the sole reliance on sampling when a single-sequence method can achieve comparable performance at a fraction of the computational cost. To ensure the scope is communicated appropriately, we have modulated this statement in the updated version of the paper (lines 537 ff. highlighted in green).
>
> > [...] only 3 datasets were considered for experiments [...]
>
> It is true that we evaluate on three datasets, which is consistent with the evaluation setup used in prior work. However, each dataset is used for generating short and long answers, resulting in six distinct tasks. For every task, we evaluate six different models, yielding **36 model–task settings** in total. This makes our evaluation more comprehensive than what is typically conducted in related work.
>
> > Also, the proposed method only works for scoring the greedy sample and not an arbitrary sample, making uncertainty estimation itself somewhat limiting. [...]
>
> It is worth noting that G-NLL is not a score of the greedy output itself. Uncertainty measures quantify how uncertain the model is about the input, rather than evaluating a particular output sequence. G-NLL uses the greedy sequence as a practical way to compute this estimate, but the resulting score reflects the model’s uncertainty about the input regardless of which output is considered.
>
> We are happy to discuss this further if any part of this would benefit from additional clarification on your side.

---

> ### Author Response · Authors · 2025-11-20
>
> Next, we would like to address the concern you raised:
>
> ---
>
> > I find the additional contribution to practice to be somewhat limited, as it seems that the measure or related measures have been explored in prior work. Also, I find the claims by the paper to be somewhat excessive – the measure is made out to be more useful than I believe it really is.
>
> It is correct that related measures have been considered as ad hoc baselines in prior work (as acknowledged in lines 57-66 of the introduction). Our main contribution, however, is a deeper insight into
> - how the underlying measure can be theoretically grounded through the MSP (Sec. 2),
> - which type of uncertainty this measure is quantifying (Sec. 2.3),
> - how its sample complexity compares to entropy-based measures (Sec. 3), and
> - how it can be approximated most effectively via G-NLL (Sec. 5.2).
>
> We further show that G-NLL outperforms computationally far more expensive uncertainty methods (Tab. 1), which have been published at top venues, such as ICLR and Nature, without considering this simple single-sequence baseline [1, 2]. **G-NLL is statistically significantly better than the best sampling-based alternative across 36 model-task settings (p = 0.0005), while requiring only a single forward pass and no hyperparameter tuning**, yet has mostly been overlooked in prior work. Even when single-sequence measures are considered, there is no consensus on how to apply them in a principled way. Our work provides this missing theoretical and practical link.
>
> ---
>
> In conclusion, we thank you again for the valuable feedback. We hope our clarifications and the updated version of the paper convey the scope and contribution of our work more clearly

---

### Official Review · Reviewer_1JBP · 2025-11-04

**Soundness:** 4
**Presentation:** 3
**Contribution:** 3
**Rating:** 8
**Confidence:** 5

**Summary:**

The paper contributes in two ways:
1. Using the framework of proper scoring rules, it shows that MSP baseline is a theoretically grounded approximation of the empirical risk.
2. More importantly, is that this approximation combined with greedy decoding might give better estimate than entropy-based scores, because the number of samples for entropy to be accurate is huge.

The authors further conduct experiments and demonstrate that fairly that MSP baseline outperforms many other methods in QA and in certain conditions. They also point out that this baselines should not be overlooked from the evaluation, which sadly happened in previous papers.

After several reviewing cycles, I believe now it is overall in a good shape and should be accepted.

**Strengths:**

1.	The authors show that MSP for greedy decoding can be shown as a relatively good approximation of the empirical risk.

**Weaknesses:**

1.	The method itself is already widely-used, though the theoretical justification was lacking before.

**Questions:**

I would like to discuss the transition to (4), could you please clarify.

---

> ### Author Response · Authors · 2025-11-20
>
> We truly appreciate your positive assessment and your conclusion that the paper is “*in good shape and should be accepted*”.
>
> In the following, we would like to address your concern and specific question raised:
>
> ---
>
> > The method itself is already widely-used, though the theoretical justification was lacking before.
>
> We agree that overlooking the MSP “*sadly happened in previous papers*”.
> Our work directly addresses this by offering the first theoretical justification for the MSP as a grounded measure of uncertainty and providing guidance for approximating it with a forward-pass.
> We show that this approximation can even outperform computationally much more expensive multi-sequence uncertainty methods (Tab. 1), which have been published at top venues, such as ICLR and Nature, without considering this simple single-sequence baseline [1, 2]. This further underscores the importance of clarifying the theoretical grounding of MSP and its practical value in closing this gap.
>
> ---
>
> > I would like to discuss the transition to (4), could you please clarify.
>
> Let us start with the posterior expectation over the expected score (line 143), where  we can add and subtract ${\color{green}{E_{p(y’|x,w)} [ S(p(y|x,w), y’) ]}}$ without loss of generality to get to Eq.(2):
>
> $E_{p(\tilde{w}|\mathcal{D})} [ E_{p(y'|x,w)} [ S(p(y | x, \tilde{w}), y’)) ] ] =
> {\color{green}{E_{p(y’|x,w)} [ S(p(y|x,w), y’) ]}} + E_{p(\tilde{w}|\mathcal{D})} [ E_{p(y'|x,w)} [ S(p(y | x, \tilde{w}), y’)) ] ]  - {\color{green}{E_{p(y’|x,w)} [ S(p(y|x,w), y’) ]}}$
>
> $E_{p(\tilde{w}|\mathcal{D})} [ E_{p(y'|x,w)} [ S(p(y | x, \tilde{w}), y’)) ] ] = {\color{green}{E_{p(y’|x,w)} [ S(p(y|x,w), y’) ]}} + E_{p(\tilde{w}|\mathcal{D})} [ {\color{green}{E_{p(y'|x,w)} [}} S(p(y | x, \tilde{w}), y’)) -  {\color{green}{S(p(y|x,w), y’) ]}} ]$
>
> $\\\\$
>
> Now, we insert the logarithmic score ${\color{olive}{S(p(y | x, \cdot), y’)) = - \log p(y'|x,\cdot)}}$ defined in Eq.(3) to get:
>
> $ E_{p(\tilde{w}|\mathcal{D})} [ E_{p(y'|x,w)} [ {\color{olive}{- \log p(y'|x,\tilde{w})}} ] ] = E_{p(y'|x,w)} [ {\color{olive}{- \log p(y'|x,w)}} ] + E_{p(\tilde{w}|\mathcal{D})} [ E_{p(y'|x,w)} [ {\color{olive}{- \log p(y'|x,\tilde{w}) + \log p(y'|x,w)}} ] ] $
>
> - On the **LHS**, we can see that $E_{p(y'|x,w)} [ - \log p(y'|x,\tilde{w}) ]$ is the definition of the cross-entropy ${\color{teal}{CE(p(y|x,w); p(y|x,\tilde{w}))}}$.
>
> - On the **RHS**, the first term $E_{p(y'|x,w)} [ - \log p(y'|x,w) ]$ is the definition of the Shannon entropy ${\color{teal}{H(p(y|x,w))}}$. The second term can be rewritten as $E_{p(y'|x,w)} [ - \log p(y'|x,\tilde{w}) + \log p(y'|x,w) ] = E_{p(y'|x,w)} [ \log \frac{p(y'|x,w)}{p(y'|x,\tilde{w})} ]$, which reveals the definition of the KL-divergence ${\color{teal}{KL( p(y|x,w) || p(y|x,\tilde{w}) )}}$.
>
> $\\\\$
>
>
> Using these definitions directly results in Eq.(4):
>
> $ E_{p(\tilde{w}|\mathcal{D})} [ {\color{teal}{CE(p(y|x,w); p(y|x,\tilde{w}))}} ] = {\color{teal}{H(p(y|x,w))}} + E_{p(\tilde{w}|\mathcal{D})} [ {\color{teal}{KL( p(y|x,w) || p(y|x,\tilde{w}))}} ]$
>
> Similarity, one can instead insert the zero-one score defined in Eq.(7), which then results in Eq.(8).
>
> $\\\\$
>
> We hope that this clarifies your question, and we are happy to provide further clarification if any points remain unclear.
>
> ---
>
> We once again thank you for the positive assessment of our work!
>
> ---
>
> [1] L. Kuhn, Y. Gal, and S. Farquhar, "Semantic uncertainty: Linguistic invariances for uncertainty estimation in natural language generation," ICLR, 2023.
>
> [2] S. Farquhar, J. Kossen, L. Kuhn, and Y. Gal, "Detecting hallucinations in large language models using semantic entropy," Nature, 2024.

---

### Author Response · Authors · 2025-12-03

Dear Reviewers and Area Chair,

Below we summarize the key points from the review period and the corresponding updates made to the paper.

---

We appreciate the positive comments that our paper “*is in a good shape and should be accepted*” (*1JBP*), that our method “*is very efficient*” (*rQPZ*), and that it overall “*is a useful contribution*” (*4Jjw*).

---

To address the raised concerns, we made the following revisions in the updated paper (highlighted in green):

- We addressed the main concerns of Reviewers *4Jjw* and *rQPZ* regarding the usefulness of our method by reporting standard errors and conducting statistical significance tests for the main experiments (Tab. 1). We show that across 36 settings covering various models and tasks, our method is statistically significantly better than the best sampling-based alternative while requiring only a single forward pass (over a 10x reduction in computational cost) and no tuned hyperparameters. This highlights the practical value of our method when applied in our theoretically grounded formulation.
- We revised and broadened the discussion of UQ (*4Jjw*) by expanding the related work section and moving it into the main paper (Sec. 4).
- We extended Eq. (9) and the surrounding discussion to clarify the section (*rQPZ*) and to explain more explicitly how G-NLL approximates MSP (*4Jjw*).
- We also clarified all remaining questions raised by the reviewers and revised the paper where appropriate.

---

Overall, we are grateful for the feedback and are confident that it helped further strengthen our work.

---

### Meta-Review · Area_Chair_53EV · 2025-12-28

**Summary:**

This paper proposes and justifies a G-NLL measure for uncertainty estimation in LLMs. The main contribution is showing the connection to an AU like term while computing the expected score for the zero-one scoring rule. As I understand, the measure itself has been explored before, but this work provides some justification. Experiments are conducted to show the merits of the simple measure over some baselines. I have reviewed a prior version of the paper and find that this revised version is extremely similar. In fact, I’m not sure about the edits that have been made in this version – they seem minor. Overall, I like aspects of this paper and think it is a useful contribution.

**Reviewer Concerns:**

1. The additional contribution to practice is somewhat limited, as it seems that the measure or related measures have been explored in prior work.
2, the author should state the method’s limitations more prominently, particularly its reliance on greedy decoding, QA-focused evaluation, and access to token-level log-probabilities.
3, the author should improve the accessibility of Sections 2–3 by adding a simple illustrative example to accompany the theoretical derivations.
4, the author should better contextualize claims about efficiency by consistently framing G-NLL as a strong baseline rather than a replacement for all sampling-based methods.

**Reviewer Scores:**

remain unchanged

---

### Decision · Program_Chairs · 2026-01-26

Accept (Poster)